# When Causal Dynamics Matter: Adapting Causal Strategies through Meta-Aware Interventions

**Moritz Willig**
Department of Computer Science
Technical University of Darmstadt

**Tim Woydt**
Department of Computer Science
Technical University of Darmstadt

**Devendra Singh Dhami**
Department of Mathematics and Computer Science
Eindhoven University of Technology

**Kristian Kersting**
Department of Computer Science
Technical University of Darmstadt
Hessian Center for AI (hessian.AI)
German Research Center for AI (DFKI)

## Abstract

Many causal inference frameworks rely on a staticity assumption, where repeated interventions are expected to yield consistent outcomes, often summarized by metrics like the Average Treatment Effect (ATE). This assumption, however, frequently fails in dynamic environments where interventions can alter the system's underlying causal structure, rendering traditional 'static' ATE insufficient or misleading. Recent works on meta-causal models (MCM) offer a promising avenue by enabling qualitative reasoning over evolving relationships. In this work, we propose a specific class of MCM with desirable properties for explicitly modeling and predicting intervention outcomes under meta-causal dynamics, together with a first method for meta-causal analysis. Through expository examples in high-impact domains of medical treatment and judicial decision-making, we highlight the severe consequences that arise when system dynamics are neglected and demonstrate the successful application of meta-causal strategies to navigate these challenges.

## 1   Introduction

Exercising agency in complex, real-world scenarios such as policy making, medical treatment, or autonomous agents carries many inherent risks and responsibilities. The consequences of actions often extend beyond immediate effects and induce lasting changes in system dynamics. Classical causal inference, largely based on structural causal models (SCMs) Spirtes et al. [2000], Pearl [2009], typically assumes a static underlying causal graph. However, this assumption often proves inadequate in dynamic settings, where causal mechanisms can evolve over time and especially under the influence of active interventions. The challenge intensifies when interventions do not only influence variables within a fixed causal structure but actively alter the causal relationships themselves. Relying on traditional metrics like direct total or Average Treatment Effect (ATE; Pearl [2009], Rubin [1980]) , which rely on the local consistency of interventional outcomes, can be misleading.

To address this, recent formalizations of Meta-Causal Models (MCM; Willig et al. [2025]) offer a promising way to model and analyze the qualitative nature of dynamical shifts in cause-effect relations. Crucially, while traditional causal analysis focuses on measuring the quantitative outcome of an intervention (e.g., average treatment effects), *Meta-Causal Analysis* (MCA) is also interested in capturing changes of the underlying transition dynamics. This allows us to answer questions that are beyond the scope of existing approaches, such as: "How likely is a system to adapt a desired state?", "How stable is a desired state?" or "Which transition paths lead to a particular state?" and

39th Conference on Neural Information Processing Systems (NeurIPS 2025).

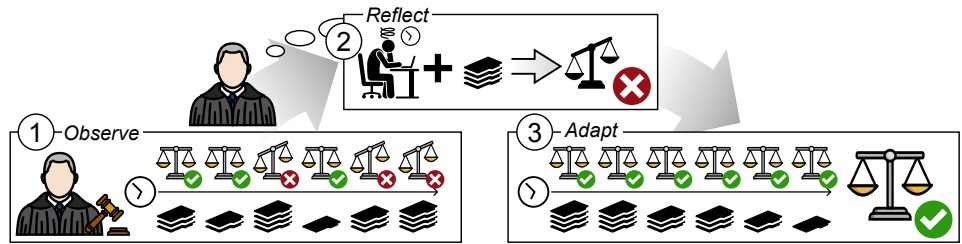

Figure 1: **Meta-Causal Adaptation in Judicial Decision Making:** Illustration of meta-causal adaptation in a judicial setting: (1) The quality of a judge's decisions degrades toward the end of sessions. (2) The judge reflects on the reasons for this decline in performance, recognizing exhaustion and complex cases as causes of increasing bias. (3) Since the judge is (hypothetically) unable to directly intervene on exhaustion levels and case complexity, the judge implements a new meta-strategy by prioritizing unclear cases earlier in the day. While the overall system dynamics remain the same, the biasing effects of fatigue are successfully mitigated and decision quality is maintained.

are these paths admissible from a safety or ethical standpoint. To the best of our knowledge, these questions cannot be answered by existing approaches, as the explicit graphical modeling of qualitative (meta-causal) state transitions is not covered and constitutes the main novelty of our paper.

**Contributions.** This paper introduces a specialized class of *Direct MCMs*, a specific assertive class of MCM that explicitly model meta-causal dynamics from within the SCM. Building on this, we formalize the first concept *Meta-Causal Analysis* (MCA), a framework for analyzing system transition between meta-causal states and proposing the *Linearized Meta-Causal Dynamics* (LMCD) algorithm to capture these dynamics. We demonstrate the critical utility of direct MCM and MCA through two examples: a medical treatment analysis showing how neglecting dynamics leads to suboptimal long-term outcomes, and a judicial decision-making setting where an agent uses meta-causal reflection to actively adapt its strategy and mitigate emerging bias.

## 2 Preliminaries

**Causal Models.** Causal relations are commonly formalized via Structural Causal Models (SCM; [Spirtes et al., 2000, Pearl, 2009]). An SCM is defined as a tuple $\mathcal{M} := (\mathbf{U}, \mathbf{V}, \mathbf{F}, P_{\mathbf{U}})$, where $\mathbf{U}$ is the set of exogenous variables, $\mathbf{V}$ is the set of endogenous variables, $\mathbf{F}$ is the set of structural equations that determine the endogenous variables, and $P_{\mathbf{U}}$ is the distribution of the exogenous variables $\mathbf{U}$. An endogenous variable $V_i \in \mathbf{V}$ is determined by a structural equation $v_i := f_i(\mathrm{Pa}(v_i))$ that takes a set of parent values $\mathrm{pa}(x_i)$, consisting of endogenous and exogenous variables that directly *cause* $V_i$, and outputs the value of $v_i$. The set of all variables is denoted by $\mathbf{X} = \mathbf{U} \cup \mathbf{V}$ with values $\mathbf{x} \in \mathcal{X}$ and $N = |\mathbf{X}|$. Every SCM $\mathcal{M}$ is associated with a (causal) graph $\mathcal{G} = (\mathbf{X}, \mathrm{E})$, that is constructed by adding edges $\mathrm{e}_{ij} \in \mathbf{X} \times \mathbf{X}$ from the causal parents $X_i \in \mathrm{Pa}(X_j)$ of each variable $X_j \in \mathbf{X}$ to itself. The *do-operator* [Pearl, 2009] written as $do(X_i = \hat{x}_i)$ alters the causal model by replacing the structural equation $f_i \in \mathbf{F}$ of the variable $X_i$ with the constant assignment $X_i := \hat{x}_i$.

**Meta-Causal Models.** *Meta-causal models* (MCM) are concerned with modeling the qualitative change of cause-effect relations in causal models over time [Willig et al., 2025]. Rather than considering specific structural equations, MCM model the qualitative causal *type* of relations between variables (e.g. 'reinforcing', 'suppressing', ...) [Chockler and Halpern, 2004, Wolff, 2007, Sloman et al., 2009, Walsh and Sloman, 2011, Gerstenberg, 2022, 2024]. To account for the factors that lead to these qualitative changes, meta-causal models assume an underlying *mediation process* $\mathcal{E} = (\mathcal{S}, \sigma)$ defined as a Markov process [Bellman, 1957] that governs the overall dynamics of the environment. The Markov process is defined over a state space $\mathcal{S}$ with a transition function $\sigma : \mathcal{S} \to \mathcal{S}$ that moves the system forward in time. MCM are typically modeled by a set of variables of interest $\mathbf{X}$, extracted from the state $\mathrm{s} \in \mathcal{S}$ of an underlying mediating process. This is done by some *abstraction* function $\varphi : \mathcal{S} \to \mathcal{X}$, which can be freely defined as a summarization or (causal) abstraction function over the state space variables [Rubenstein et al., 2017, Beckers and Halpern, 2019, Anand et al., 2022, Wahl et al., 2023, Kekić et al., 2023, Willig et al., 2023]. The linkage between a Markov process and a causal model is formally captured via *Meta-Causal Frames* (Def. 1). Each *meta-causal state* (MCS; Def. 2) is defined as $T \in \mathcal{T}^{N \times N}$ captures the current qualitative type of relationship between all pairs

of causal variables at a given environment state. Each element $t_{ij} \in \mathcal{T}$ indicates the particular *type* of relationship between any two variables $X_i, X_j$. The special type $0 \in \mathcal{T}$, indicates the complete absence of an edge. Thus, MCS are a generalization of a causal model's adjacency matrix. Given a mediation process, meta-causal frame emitting meta-causal states, MCM (Def. 3) model the change of functional dependencies for different states of the environment and thus capture the transition dynamics between different configurations of the causal graph. We briefly repeat the necessary definitions of MCM for further consideration in the following sections:

**Definition 1** (Meta-Causal Frame; MCF). *For a given mediation process $\mathcal{E} = (\mathcal{S}, \sigma)$ a **meta-causal frame** is a tuple $\mathcal{F} = (\mathcal{E}, \mathbf{X}, (\tau_{ij}), \mathcal{I})$. **Type-encoders** $\tau_{ij} : \mathcal{X}_i \times \boldsymbol{\mathcal{X}}^{\mathcal{S}} \to \mathcal{T}$ assign a **type** $t \in \mathcal{T}$ to the functional dependency of $X_j$ on $X_i$, induced by the underlying process $\mathcal{E}$, which is a relation between $\mathcal{X}_i$ (values of $X_i$) and the abstraction of the transition function $\varphi \circ \sigma \in \boldsymbol{\mathcal{X}}^{\mathcal{S}} = \{\psi : \mathcal{S} \to \boldsymbol{\mathcal{X}}\}$. The **identification function** $\mathcal{I} : \mathcal{S} \times \mathbf{X} \times \mathbf{X} \to \mathcal{T}$ with $\mathcal{I}(s, X_i, X_j) \mapsto t := \tau_{ij}(\varphi(s), \varphi \circ \sigma)$ assigns a type to each pair of causal variables for each state of the environment.*

Within a Meta-Causal Frame every system state $s \in \mathcal{S}$ is mapped to a meta-causal state $T \in \mathcal{T}^{N \times N}$ that captures the current type of relations between all variables:

**Definition 2** (Meta-Causal State; MCS). *In a meta-causal frame $\mathcal{F} = (\mathcal{E}, \mathbf{X}, (\tau_{ij}), \mathcal{I})$, a **meta-causal state** is a matrix $T \in \mathcal{T}^{N \times N}$. For a given environment state $s \in \mathcal{S}$, the **actual meta-causal state** $T_s$ has the entries $T_{s,ij} := \mathcal{I}(s, X_i, X_j) = \tau_{ij}(\varphi(s), \varphi \circ \sigma)$.*

Finally, a meta-causal model identifies the meta-causal dynamics from the current state of the system $s \in \mathcal{S}$ and the state transitions of the Markov process $\sigma$ at every point in time. The dynamic changes in the mediation process are thus observed as qualitative changes in the meta-causal state:

**Definition 3** (Meta-Causal Model; MCM). *For a meta-causal frame $\mathcal{F} = ((\mathcal{S}, \sigma), \mathbf{X}, (\tau_{ij}), \mathcal{I})$, a **meta-causal model** is a finite-state machine defined as a tuple $\mathcal{A} = (\mathcal{T}^{N \times N}, \mathcal{S}, \delta)$, where the set of meta-causal states $\mathcal{T}^{N \times N}$ is the set of machine states, the set of environment states $\mathcal{S}$ is the input alphabet, and $\delta : \mathcal{T}^{N \times N} \times \mathcal{S} \to \mathcal{T}^{N \times N}$ is a transition function consistent with the environment transition $\sigma$ and type encoders $\tau_{ij}$.*

## 3 Predicting Meta-Causal Change

One of the main hurdles that has made it difficult to transfer decisions from general meta-causal models back to the underlying Markov process has been the abstraction that relates the Markov process to the causal variables. In this section, we consider a particular class of meta-causal models that link the two models more tightly and, in turn, are more informative in terms of feedback. First, we introduce a class of MCM that directly considers the variables of the underlying mediation process as their causal variables. Second, we show that meta-causal state transitions become directly predictable from the causal model under a certain choice of abstraction functions. In particular, we identify a set of *meta-causal variables* that are responsible for these transitions. Finally, we discuss context dependencies as a special case of meta-causal dynamics, relevant to the later Applications section.

**Direct MCM.** We consider a class of MCM where the variables of the underlying mediation process $\mathcal{S}$ and the causal model $\mathbf{X}$ are considered to model the same set of variables:

**Definition 4** (Direct MCM). *For a meta-causal frame $\mathcal{F} = ((\mathcal{S}, \sigma), \mathbf{X}, (\tau_{ij}), \mathcal{I})$ a meta-causal model $\mathcal{A} = (\mathcal{T}^{N \times N}, \mathcal{S}, \delta)$ is called **direct meta-causal model** if $\mathcal{S} = \boldsymbol{\mathcal{X}}$ and $\varphi = \mathrm{Id}$.*

Since meta-causal states are governed by the transition function of the mediating process $\sigma : \mathcal{S} \to \mathcal{S}$, the structural equations are now also defined in terms of the state transitions $\mathbf{F} := \varphi \circ \sigma \circ \varphi^{-1} = \mathrm{Id} \circ \sigma \circ \mathrm{Id}^{-1} = \sigma$. In the context of meta-causal models (or dynamical systems in general), the mediating process $\sigma$ factorizes into two sets of equations governing either the causal variables within a given particular time step or meta-causal state $X_t \to Y_t$, which can be denoted as $\sigma^{t \to t}$, and those that transition the system to the next time step (or meta-causal state) $X_t \to Y_{t+1}, \sigma^{t \to t+1}$.

### 3.1 Factors of Meta-Causal Change

Previous work on meta-causal models either had no explicit notion of factors that caused changes in the MCM, or attributed these changes either to explicit interventions or to unspecified (exogenous)

environmental factors [Minka and Winn, 2008, Peters et al., 2016, Seitzer et al., 2021, Willig et al., 2025]. In this paper, we assume that the factors that have the ability to change the (meta-causal) type of relations are observed within the model, and thus can be identified as a subset of the variables. A key aspect in distinguishing ordinary variables from such 'meta-causal' variables is their ability to influence the qualitative type $T_{ij}$ of any relation within the model. The notion of what constitutes a meta-causal variable also includes the currently used identification function $\mathcal{I} : \mathcal{S} \times \mathbf{X} \times \mathbf{X} \to \mathcal{T}$, since it ultimately determines what constitutes a change in meta-causal types in the first place. Similar to how parents in the standard SCM are sometimes characterized by their ability to influence the values of their respective child variables, we formalize *meta-causal variables* (MCV) as the set of variables that have the potential to change the identified *type* of one or more causal relations within the SCM. (We write $\mathbf{x}_{\bar{k}}$ to mean the vector without the $k$-th entry.)

$$\mathbf{C} := \{ X_k \in \mathbf{X} \mid \exists X_i, X_j \in \mathbf{X} . \exists \mathbf{x}, \mathbf{x}' \in \mathcal{X} \text{ s.t.} \tag{1}$$
$$(\mathbf{x}_{\bar{k}} = \mathbf{x}'_{\bar{k}}) \wedge (x_k \neq x'_k) \wedge (\mathcal{I}(\mathbf{x}, X_i, X_j) \neq \mathcal{I}(\mathbf{x}', X_i, X_j))\}$$

MCV are a subset of the standard variables ($\mathbf{C} \subseteq \mathbf{X}$). Most closely related to the notion of MCVs is the work of Günther et al. [2024], which is concerned with discovering context variables from data, but does not further model their associated dynamics. A notion of MCVs was also implicitly used in the examples of Willig et al. [2025]. Here we provide a first explicit definition of MCV.

**Predictability of Meta-Causal Dynamics.** The ability to make predictions about causal dynamics from observations of a system is a core aspect of meta-causal models. Here, MCV specifically capture the relevant factors governing the dynamics of the system. While other exogenous factors may influence the course of meta-causal evolution, we are interested in systems that can be predicted from within the SCM. We formalize this form of predictability as follows:

**Definition 5** (Meta-Causal Predictability). *A Meta Causal Model* $(\mathcal{T}^{N \times N}, \mathcal{S}, \delta)$ *is called* **meta-causal predictable** *if the next meta-causal state* $T_{s_{t+1}} \in \mathcal{T}^{N \times N}$ *can be predicted purely from the variable values at the current time step* $\mathbf{x}_t \in \mathcal{X} = \mathcal{S}$, *so that the transition function takes the form* $\delta^{\mathcal{X}} : \mathcal{X} \to \mathcal{T}^{N \times N}$.

State transitions may not need to be fully deterministic. The above definition can be reformulated into a probabilistic variant where $\delta^{\mathcal{X}} : \mathcal{X} \to \mathcal{P}(\mathcal{T}^{N \times N})$ is a probability measure over the meta-causal states $\mathcal{T}^{N \times N}$. Since transition probabilities can shift depending on the current meta-causal state, a relaxed notion of meta-causal predictability might reintroduce the current MCS $T_{s_t}$ as a parameter, $\delta^{\mathcal{X}} : \mathcal{X} \times \mathcal{T}^{N \times N} \to \mathcal{P}(\mathcal{T}^{N \times N})$. The predictability of a system can then be measured using Shannon entropy ($\mathbf{H}(\mathcal{P}) := -\sum_{p_i \in \mathcal{P}} p_i \log(p_i)$; Shannon [1948]). The predictability of a system then increases with increasing entropy, $\mathbf{H}(\mathcal{P}(\mathcal{T}^{N \times N})) \to 1$. For strict meta-causal predictability ($\mathbf{H}(\mathcal{P}(\mathcal{T}^{N \times N})) = 1$), $\mathcal{P}(\mathcal{T}^{N \times N})$ must then be a point-mass distribution, with all meta-causal state transitions being deterministic and fully governed by the set of meta-causal variables $\mathbf{C}$.

For arbitrary abstractions $\varphi$ this cannot be guaranteed, since relevant information about the factors responsible for state transitions may be marginalized by the abstraction. The choice for direct MCM (Def. 4) to set $\varphi$ as identity provides a particularly straightforward way to guarantee meta-causal predictability. In general, any abstraction function that preserves full information about the underlying transition factors (e.g., $\varphi$ being bijective w.r.t. $\mathbf{C}$) is suitable for preserving meta-causal predictability.

**Theorem 3.1.** *Every MCM* $(\mathcal{T}^{N \times N}, \mathcal{S}, \delta)$ *with a bijective abstraction* $\varphi$ *is meta-causal predictable.*

*Proof.* Since $\varphi$ is bijective, its inverse $\varphi^{-1}$ exists and is invertible with $\mathrm{s} = \varphi^{-1}(\mathbf{x})$. The assignment $\mathbf{F} := \varphi \circ \sigma \circ \varphi^{-1}$ implies $\sigma = \varphi^{-1} \circ \mathbf{F} \circ \varphi$. By Def. 2, types are determined by $T_{\mathrm{s},ij} := \tau_{ij}(\varphi(\mathrm{s}), \varphi \circ \sigma)$ and $T_{\mathrm{s}+1,ij} := \tau_{ij}(\sigma(\mathrm{s}), \varphi \circ \sigma) = \tau_{ij}((\varphi^{-1} \circ \mathbf{F} \circ \varphi)(\varphi^{-1}(\mathbf{x})), \varphi \circ (\varphi^{-1} \circ \mathbf{F} \circ \varphi)) = \tau_{ij}(\varphi^{-1} \circ \mathbf{F}(\mathbf{x})), \mathbf{F} \circ \varphi)$, which is a function $\delta^{\mathcal{X}} : \mathcal{X} \to \mathcal{T}^{N \times N}$ that is completely governed by the SCM's $\mathbf{x}$ and $\mathbf{F}$. $\square$

The bijectiveness condition of $\varphi$ might be relaxed further. Implications are discussed in App. A. Meta-causal predictability of direct MCM follows directly from Thm. 3.1 and the identity being bijective. Assuming bijective abstraction functions, the state transition $\sigma : \mathcal{S} \to \mathcal{S}$ can be defined in terms of causal variables $\sigma^{\mathcal{X}} : \mathcal{X} \to \mathcal{X}$ with $\sigma^{\mathcal{X}} := \varphi \circ \sigma \circ \varphi^{-1}$, (in particular $\sigma^{\mathcal{X}} := \mathrm{id} \circ \sigma \circ \mathrm{id}^{-1} = \sigma$ for direct MCM). As a result, the overall MCM transitions $\delta : \mathcal{T}^{N \times N} \times \mathcal{S} \to \mathcal{T}^{N \times N}$ can also be written

in terms of causal variables $\delta^{\boldsymbol{\mathcal{X}}} : \boldsymbol{\mathcal{X}} \rightarrow \mathcal{T}^{N \times N}$ with $\delta^{\boldsymbol{\mathcal{X}}}(\mathbf{x}) := \mathcal{I}(\sigma^{\boldsymbol{\mathcal{X}}}(\mathbf{x}), X_i, X_j) = \tau_{ij}(\sigma^{\boldsymbol{\mathcal{X}}}(\mathbf{x}), \mathbf{F})$ similar to the original definition with $\sigma$ swapped.

**Contextual Independencies and Discovery of MCM.** As a final part of this section, we briefly discuss the discovery of MCVs of MCM for the special class of contextually switching MCM. From a meta-causal perspective, an important class of switching causal mechanisms is that of contextual independencies, where relations either exert a unique type $t_{ij}^*$ while being active or are (contextually) independent otherwise, $T_{ij} \in \{t_{ij}^*, 0\}$. Identifying particular context variables that lead to switching causal graphs has been the subject of several approaches in general causal discovery [Pensar et al., 2015, Hyttinen et al., 2018, Günther et al., 2024], reinforcement learning, and general prediction of system dynamics [Seitzer et al., 2021, Liu et al., 2023]. While some works dealing with switching dynamics do not have an explicit notion of such context variables [Seitzer et al., 2021, Liu et al., 2023], several others deal with their discovery from data (but without modeling dynamics) under the name of 'Labeled Directed Acyclic Graphs' (LDAGs; Pensar et al. [2015], Hyttinen et al. [2018], Günther et al. [2024]). In particular, [Günther et al., 2024] relaxes the previous assumption to discover contextual graphs from shared data. Under the previous assumption of meta-causal predictability and the discovery of observed endogenous context variables (or more general MCVs), the general dynamics of MCM state transitions can be recovered. In this work we are not concerned with discovery, but use the notion of contextual independency as meta-causal phenomena in our examples in Sec. 6. This concludes our formal considerations of the predictability of MCM dynamics.

# 4 Intervention Effects on Meta-Causal Stability

Causal modeling is commonly used to support domain understanding and subsequent decision making. With meta-causality being the main focus of this paper, we consider scenarios where decisions induce lasting changes to the system that have the ability to permanently alter the system dynamics. While the previous section discussed conditions for meta-causal predictability, we will now draw attention to the insights that can be gained from such models. Drawing qualitative inferences not only for individual decisions, but also for strategies that continuously affect a system, is a key capability for making reliable and robust decisions [Bareinboim et al., 2021, Zhang and Bareinboim, 2022, Aalaila et al., 2025]. Discussions of the role of long- versus short-term outcomes [Lear and Zhang, 2025] and the adaptation of reflective strategies have been discussed in previous work [Lee and Bareinboim, 2018, Dasgupta et al., 2019, Boeken et al., 2024]. In this section, we focus on describing those qualitative changes in system dynamics with the help of meta-causal considerations.

To study the influence of one variable on another, the (direct) causal effect –as for example measured by the *Average Treatment Effect* (ATE; Pearl [2009], Rubin [1980])– is usually approximated from a set of individual samples $(x_i, y_i)_{i \in [1..n]}$. Within the potential outcomes framework (under the assumptions of exogeneity and ignorability), the ATE is defined as the expected difference in outcome that would result from treating an individual $i$ ($y_1(i)$) compared to not treating them ($y_0(i)$):

$$\text{ATE} = \mathbb{E}[y_1 - y_0] \approx \frac{1}{n} \sum_i y_1(i) - y_0(i) \tag{2}$$

In the following discussion we make use of the *do*-operator ($Y^{do(X_i=a)}$) –where $a \in \{0, 1\}$ indicates the presence or absence of a treatment, respectively– to distinguish the interventional treatment setting from simple conditioning, which may be susceptible to spurious confounding:

$$\text{ATE} = \mathbb{E}[y^{do(X=1)} - y^{do(X=0)}] \approx \frac{1}{n} \sum_i y_i^{do(X_i=1)} - y_i^{do(X_i=0)} \tag{3}$$

Since we base our formalism in the Pearlian causal framework, the Pearlian do-calculus states a set of exact and complete conditions under which causal effects can be identified for arbitrary graphical structures and beyond the bivariate case [Pearl, 1994, Tian and Pearl, 2002, Shpitser and Pearl, 2006, Huang and Valtorta, 2006]. For ease of notation, and since variable adjustment is not the focus of this paper, we only consider cases where the above Eq. 3 directly captures unbiased estimates of the causal effects to be estimated from $X$ on $Y$ without further consideration of possible adjustment sets.

**Stability of Meta-Causal Dynamics.** These considerations of causal effects are made under various assumptions, such as the Stable Unit Treatment Value Assumption [Rubin, 1974, 1980], which assumes that the treatment of one participant generally has no effect on the intervention outcome of others. Furthermore, most systems are assumed to be well-behaved in the sense that their overall local

dynamics remain stable regardless of the intervention being studied [Kevorkian, 1966, Kevorkian and Cole, 1968]. Assumptions on the stationarity of causal dynamics can be conveniently formalized from a meta-causal perspective. One way to express the local stability of the system under perturbation is to assume that the system always remains in its current meta-causal state, regardless of the type of intervention applied. Assuming meta-causal predictability (Assm. 5), we take advantage of the fact that the MCS can be inferred from the causal variables $\mathbf{x} \in \mathbf{X}$, and write $T_{\mathbf{x}}$ and $T_{\mathbf{x}|do(\mathbf{y})}$ to denote the meta-causal states arising from $\mathbf{x}$ and from $\mathbf{x}$ under the interventions $do(\mathbf{y})$, respectively. In its most conservative form, *strict meta-stability* is formalized as follows:

$$\forall \mathbf{x}, \mathbf{x}' \in \mathbf{X} . \forall \mathbf{y} \subseteq \mathbf{x}' . T_{\mathbf{x}|do(\mathbf{y})} = T_{\mathbf{x}} \tag{4}$$

Under the criterion of Eq. 4, the meta-causal state must not be altered at any point in time, including points of intervention. However, this strict notion may preclude the common use of hard interventions. As hard interventions cut the edges from the intervened variable $X_i$ to its parents, it in turn changes the meta-causal types of these edges from their previous values to zero-type ($0$). This, alters the meta-causal state $T_{\mathbf{x}|do(\mathbf{y})} \neq T_{\mathbf{x}}$ so that these actions are not allowed. Under the strict notion of Eq. 4, either soft interventions or interventions on root nodes (e.g., instrumental variables) are required, which either do not alter the meta-causal type or do not cut any edges. Depending on the use case, a weaker criterion may be sufficient, requiring only that the system converges back to its initial state some time after the intervention. Superscripts indicate the meta-causal state at particular points in time, as well as the time of the intervention:

$$\forall \mathbf{x}, \mathbf{x}' \in \mathbf{X} . \forall \mathbf{y} \subseteq \mathbf{x}' . \exists t, \Delta t \in \mathbb{N}^+ . (t < t' \wedge \exists do^t(\mathbf{y})) \Rightarrow (T_{\mathbf{x}}^{t-1} = T_{\mathbf{x}}^{t+\Delta t}) \tag{5}$$

The relaxed notation of Eq. 5 makes no statement about the MCS during the time of intervention, but only requires that the system converges back to its initial state after some time $\Delta t$. This allows in particular the use of hard interventions, as long as their effects are reversed after some time.

**Stability of Stationary SCM.** Stationary SCM without time-dependent system dynamics ($\sigma^{t \rightarrow t+1} = \emptyset$) trivially satisfy Eq. 5. Since each $\mathbf{x}^t$ depends only on its current (independently sampled) noise at time $t \in \mathbb{N}^+$, so does $T_{\mathbf{x}}^t$. Any effects of an intervention $do^t(\mathbf{y})$ vanish at the end of the intervention, since no information is carried over from $\sigma^{t \rightarrow t+1}$. As a result, $\Delta t$ can be set to $\Delta t := 1$, (assuming no subsequent interventions change the MCS $T_{\mathbf{x}}^{t+1}$ at the following timestep). The unsatisfiability of Eqs. 4 and 5, serve as indicators that meta-causal dynamics might have been permanently altered following some interventions. Qualitative changes in mechanisms do not always come as sudden, abrupt changes, but can be the consequence of a series of repeated interventions that steadily affect the system's dynamics until some tipping point is reached (recognized as a change in $T$ via the meta-causal identification function $\mathcal{I}$). To avoid unwanted outcomes, it is of primary importance to consider not only the short-term effects of some actions but also the qualitative long-term consequences of shifted dynamics. In the following section we propose a way of analyzing such meta-causal dynamics.

## 5 Meta-Causal Analyses

Meta-Causal Analysis (MCA) focuses on capturing changes in the underlying meta-causal state transition dynamics, offering a perspective distinct from existing analysis methods, such as Dynamic Treatment Regimes (DTR) or longitudinal statistics, which primarily measure the eventually emitted outcome effects of interventions. Questions amenable to MCA include determining the likelihood of a system adapting a desired meta-causal state, assessing the stability of a state, or identifying the transition pathways (sequences of meta-causal states) available to reach a particular state. As transition dynamics eventually affect actual long term outcomes, MCA is intended to supplement existing methods by explicitly modeling the flow between system configurations with the help of graphical causal models, rather than only observing resulting outcomes.

**Linearized Meta-Causal Dynamics Algorithm.** To facilitate capturing meta-causal effects, we propose the *Linearized Meta-Causal Dynamics algorithm* (LMCD; Alg. 1), which captures the linearized dynamics –meaning, that no second-order effects, e.g., shifts in time, on the observed transition probabilities are assumed to be present– by approximating transition probabilities between meta-causal states from a sample population $[\mathbf{x}^1, \ldots, \mathbf{x}^N] \in \mathbf{X}^N, N \in \mathbb{N}$ and a meta-causal predictable model. Under the assumption of meta-causal predictability (Assm. 5) the structural equations are sufficient to advance the system onto the next state (Thm. 3.1). To capture the true transition probabilities, we furthermore assume that the presented data is sampled identically to the underlying

---

**Algorithm 1** Linearized Meta-Causal Dynamics (LMCD) Algorithm

---

1: **Input:** SCM: $\mathcal{M} = (\mathbf{V}, \mathbf{U}, \mathbf{F}, P_{\mathbf{U}})$, data: $\mathbf{x^I} = (\mathbf{x}^i)_{i=1}^N \in \mathbf{X}^N$, id. func.: $\mathcal{I} : \mathbf{X} \to \mathrm{T}$
2: **for each** $\mathbf{x}^i$ in $\mathbf{x^I}$ **do**
3:      $\mathbf{x}^{i,t+1} \leftarrow \mathbf{F}((\mathbf{x}^i |_{\mathbf{V}}) \cup (\mathbf{u}^{t+1} \sim P_{\mathbf{U}}))$               ▷ Advance the system.
4:      $(\mathrm{T}^{i,t}, \mathrm{T}^{i,t+1}) \leftarrow (\mathcal{I}(\mathbf{x}^i), \mathcal{I}(\mathbf{x}^{i,t+1}))$          ▷ Identify MCS transition pair.
5: $U \leftarrow (\bigcup_i l(\mathrm{T}^{i,t})) \cup (\bigcup_i l(\mathrm{T}^{i,t+1}))$          ▷ Determine set of unique MCS.
6: **for each** $(u, v)$ in $\{1, \ldots, |U|\}^2$ **do**      ▷ Approximate transition dynamics, $P \in \mathbb{R}^{|U| \times |U|}$.
7:      $P_{u,v} \leftarrow \sum_{i \in [1..N]} (\mathbf{1}((l(\mathrm{T}^{i,t}) = u) \wedge (l(\mathrm{T}^{i,t+1}) = v))) / \sum_{i \in [1..N]} \mathbf{1}(l(\mathrm{T}^{i,t}) = v))$
8: $[Q \leftarrow e^{P-I}]$      ▷ Optional: Compute continuous time rate matrix. ($I$ is the identity matrix.)
9: **return** $P, [Q]$

---

MCS distribution. Instead of advancing single data points, pairs of consecutive data points could directly be considered, if available. This would relate the LMCD method more closely to classical time-series analysis and eliminates the meta-causal predictability assumption.

In a first step, all given sample points are advanced in time (indicated by '$t + 1$'). Next, the set of actually reachable MCS $U$ is computed. For practical purposes, an indexing function $l : T \to [1..|T|]$ is introduced that assigns a unique index to each MCS $t \in T$. This function can equally be used to group different $T$ by only considering a problem relevant subset $T' \subseteq T$ of the full type matrix when assigning the index. Finally, transition probabilities $P \in \mathbb{R}^{|U| \times |U|}$ are computed. ($\mathbf{1}(b)$ is the indicator function which is 1 if $b$ is true and 0 otherwise.) A time rate matrix $Q$ can be determined for continuous time setups.

**Meta-Causal ATE.** The *(specific) Meta-Causal ATE* (sMCATE) between two strategies $A$, $B$ with transition matrices $P_A$, $P_B$ (or optionally $Q_A$, $Q_B$) might be defined as the difference in their transition probabilities $\text{sMCATE}(P_A, P_B) := P_B - P_A$. Note, that $P_A, P_B$ already approximate the transition expectations over the respective populations. In line with the discussions in Sec. 4 it is assumed that transition matrices $P_A, P_B$ are able to capture the underlying system dynamics and that those do not shift due to higher-order effects. The sMCATE might further be compressed into a single number via the use of matrix norms for specific scenarios. In the absence of a clear candidate for defining a general MCATE we abstain from deciding on a definite characterization.

## 6 Applications

After addressing the formal aspects of predictability and stability, we now turn to illustrative use cases of meta-causality. Sec. 6.1 compares static and dynamic ATE analyses in a flu medication example, highlighting their differing implications from a meta-causal perspective. Then, Sec. 6.2 examines judicial decision-making, where agent fatigue introduces bias. Here, a meta-aware agent can recognize and counteract such biases through meta-level intervention. Code for reproducing examples is provided at: https://github.com/MoritzWillig/metaCausalDynamics.

For the examples, the structural equations within the SCM do not switch. To identify whether parents have an actual influence on their children, we use an identification function that determines the presence of an edge by inspecting whether the connection structural equations have a non-zero gradient. We refer to App. B for more details.

### 6.1 Medicating Flu

In this example, we revisit the discussion of short- versus long-term effects [Lear and Zhang, 2025] with a particular focus on the stability of meta-causal states. The example considers the effects of different drugs used to treat fever. We consider a strongly simplified system with exaggerated system dynamics for clarity of the example. Similarly, we deem both medications sufficient for treating all levels of fever, such that no trade offs between short- and long-term outcomes are considered here.

At each time step, participants experience varying levels of viral exposure ($E_t \in \mathbb{R}^+$). Fever occurs when the viral load exceeds the immune system's current capacity $F_t := max(E_{t-1} - I_t, 0)$ and drugs are actively given in consequence, $M_t := \mathbf{1}(F_t > 0.5)$. Drugs $M_k$, $k \in \{A,B\}$ are characterized

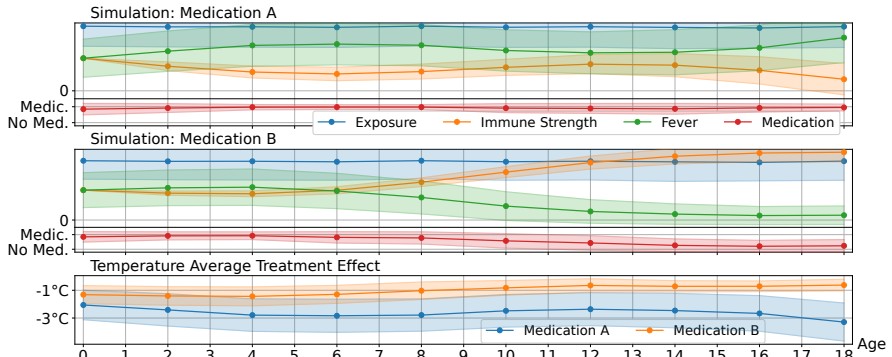

Figure 2: **Medicating Flu.** The top and center plots show the averaged simulated evolution of the system under different drugs. Shaded areas show standard deviations. Medication A provides a consistently stronger direct reduction in body temperature according to the ATE shown in the bottom plot. However, it also suppresses immune development to the point where each subsequent viral exposure requires treatment. Medication B has a weaker direct ATE, but does not inhibit the immune system, resulting in a significant long-term reduction in fever.

by their parameters $(\alpha_{M_k}, \beta_{M_k})$), which differ in suppressing fever symptoms $(\alpha_{M_k})$ of rising body temperature $(B_t := F_t \cdot (1 - \alpha_{M_k} M_t))$, but could also impede the development of the immune system that would otherwise develop naturally over time $(I_t := I_{t-1} + \text{sigm}(t) - \beta_{M_k} M_{t-1})$; where sigm is a sigmoid function representing immune maturation). The corresponding causal graph is shown in Fig. 3. Full equations and starting conditions are given in App. C. We consider two drugs, with arbitrary chosen parameters: Drug A has a high immediate suppressive effect on fever symptoms, but also a stronger negative effect on the immune system, $M_A = \{\alpha_A = 0.95; \beta_A = 0.75\}$. Drug B has a milder fever response, but also a milder effect on the immune system, $M_B = \{\alpha_B = 0.6; \beta_B = 0.4\}$.

We performed simulations over 1,000 repetitions (details given in App. C). Fig. 2 (top and center) shows the average evolution of the system under drugs A and B over time. To assess the efficacy of either drug, a conventional causal analysis might measure the 'direct' ATE in terms of the immediate reduction in body temperature. We analyze how the different medications affect the qualitative system dynamics. The result is summarized in the bottom plot of Fig. 2 as the average difference in body temperature between treating and not treating a fever with each drug, $\text{ATE}(M_t) := \mathbb{E}[B_t^{do(M_t=1)} - B_t^{do(M_t=0)}]$. As can be clearly seen from the figure, drug A is more effective at lowering the temperature over all time steps (with an overall average standard deviation of $\text{ATE}^{M_A} = -2.61°C \pm 0.34°C$) than drug B ($\text{ATE}^{M_B} = -1.26°C \pm 0.78°C$). An analysis based on this quantity alone would therefore favor drug A as the superior treatment.

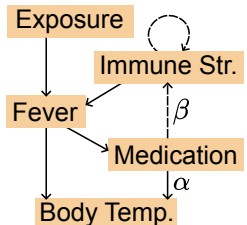

Figure 3: **Medicating Flu:** Causal graph for viral exposure and immune response under the influence of medication. Solid lines indicate instantaneous relationships; dashed lines indicate relationships between successive time steps.

From a Meta-Causal Analysis (MCA) perspective, drug A's cumulative negative effect on the immune system is evident by analyzing the transition dynamics between meta-causal states (MCS). Focusing on the Fever → Temperature and Medication → Temperature edges via the indexing function $l$ of the LMCD algorithm (Alg. 1), transition probabilities between the three MCS "no fever (no treatment)", "mild fever (no treatment)", and "fever with medication" –there is no treatment without fever– are approximated. Applying LMCD to the simulation data at age 18 shows that drug A leads to a high probability (99.39%) of remaining in the medication-required MCS, with all other states having a high probability (up to 77.78%) of flowing into this state, indicating an induced dependency on medication. In contrast, drug B's dynamics are more favorable, exhibiting a lower chance for recovery-related state transitions and a reduced probability of remaining in the medication-required MCS (61.22%). The sMCATE quantifies this difference, showing that drug B strongly decreases the likelihood of transitioning into (reduction up to 36.43pp) or remaining in the dependency-inducing state ($-38.18$pp). This analysis

suggests preferring drug B (if sufficient for treatment). A fully worked example, showing transition matrices and sMCATE computations is provided in App. E.

## 6.2 Judicial Decision-Making

In this example we present the case of a meta-aware agent that actively reflects on the system dynamics and is therefore able to adapt its strategy to counteract emerging biases. Similar considerations of algorithmic recourse and dynamic treatment Zhang and Bareinboim [2019], Zhang [2020], Gerstenberg [2022], Von Kügelgen et al. [2022] have already been considered for classical causal cases. Here, we put our focus on the meta-causal aspects which include the agent actively self-intervening on its own policy function when becoming aware of the arising bias.

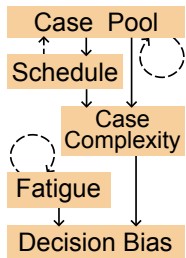

An overview of the scenario is illustrated in Fig. 1, with the corresponding causal graph presented in Fig. 4. Full equations and starting conditions are given in App. D. In this scenario, a judge hears $N = 6$ cases per day, drawn from a given pool of daily cases $p_0 = \{c_1, \ldots, c_N\}$, where each case is assigned a complexity, $c_i \in \{1, 2, 3, 4\}$, that is a proxy for the amount of documents to read or the criticality of the case (e.g., the maximum sentence to expect). For each time slot, the judge selects one of the cases according to their schedule $S : \{\mathbb{R}^+\} \to [1..N]$ with $s_t := f(p_t)$, from which the current case complexity $cx_t := p_{S_t}$ is determined and which can't be heard again afterwards, $p_t := p_{t-1} \setminus \{s_t\}$. Over the course of the day, the judge gradually becomes fatigued, $f_t := f_{t-1} + 0.5$, which affects the decisions made, so that when the judge encounters complex cases while fatigued, the decisions become biased $b_t := max(f_t + cx_t - 5, 0)$.

Figure 4: **Judicial Decision-Making:** The corresponding SCM for the Judicial Decision-Making setting. Solid lines indicate instantaneous relations; dashed lines indicate relations between consequent time steps.

Reflecting on past decisions, the judge notices an emerging fatigue-induced bias when hearing complex cases. Identifying fatigue and case complexity as the direct causes of the bias, the judge considers ways to counteract its effects. Considering actual edge activations via the previously described identification function (App. B), a suitable strategy has to avoid any edge activations of fatigue $\to$ decision bias and case complexity $\to$ decision bias. Since the emergence of decision bias is not under the direct control of the judge, the only way to avoid bias is to steer the system dynamics into a state $\mathrm{T}^*$ where, at any point in time, neither $F_t \to B_t$ nor $Cx_t \to B_t$ exert causal influence; $\mathrm{T}^* \Rightarrow ((\mathrm{T}_{F_t, B_t} = 0) \vee (\mathrm{T}_{Cx_t, B_t} = 0))$. Through the additive, thresholded interplay of fatigue and case complexity, both factors act as meta-causal variables to each other, in the sense that low fatigue prevents case complexity from exerting influence on the occurrence of bias, and vice versa. Fatigue levels are assumed to be unintervenable and steadily rising, such that during later hearings, the edges have the potential to become active and the meta-causal type is solely dependent on case complexity. Conversely, complex cases that might lead to biased decisions might be best heard in earlier sessions when the judge's concentration is high (implying low fatigue). As a result of this, the judge adopts a new scheduling strategy $do(S := f^*)$ with $f^*(p) = \operatorname{argmax}_i(\{p_i\}_{p_i \in p})$, which schedules complex cases during periods of low fatigue, thus preventing them from biasing decisions in later hearings. Under the new strategy, the system dynamics remain within the desired state $\mathrm{T}^*$.

The previous example shows how meta-causal reasoning can prevent fatigue-induced bias by reflecting on system dynamics. While similar insights may arise from earlier methods, meta-causality defines explicit conditions for edge activation, enabling logical strategy adaptation. A full comparison of initial and adjusted policy dynamics is provided in App. F.

## 7 Related Work

**Longitudinal Statistics.** Works by Robins [1986, 1997] on the g-formula aim to estimate exposure effects in the presence of time-varying confounders. Most similar to our work, Robins et al. [2000] leverages marginal structural models for causal effect estimation under time-dependent confounding. Liang and Zeger [1986] and Laird and Ware [1982] utilize linearized models and random-effect sampling to approximate the effects of long-term dynamic shifts in the resulting distributions and corresponding ATE. In contrast to prior work, which primarily focuses on measuring the actual emitted effects, e.g., in dynamic treatment regimes [Murphy, 2003], MCA is designed to supplement

existing methods by explicitly considering transition dynamics and pathways between different qualitative meta-causal states. It is therefore not only interested in the resulting outcome effects, but also how the resulting effects emerge within the system.

**Time Series and System Dynamics.** The study of dynamical systems in causality is a long-standing field of study [Friston et al., 2003, Mooij et al., 2013, Bongers et al., 2018, Blom et al., 2020, Peters et al., 2022, Löwe et al., 2022, Lear and Zhang, 2025] with common application in the modeling of climate systems [Zscheischler et al., 2020, Camps-Valls et al., 2023, Runge et al., 2023] or general time-lagged relations [Peters et al., 2013, Saggioro et al., 2020, Runge et al., 2019, Gerhardus et al., 2023, Runge et al., 2023]. The general modeling and unsupervised learning of such systems under causal aspects has also been widely studied [Dash, 2005, Hyttinen et al., 2012, Mooij et al., 2013, Hansen and Sokol, 2014, Chalupka et al., 2016, Rubenstein et al., 2016, Bongers et al., 2021, Peters et al., 2022, Blom and Mooij, 2023]. In contrast to our work, most prior work either consider models with stationary equations or does not explicitly model factors that cause changing structural equations.

**Switching Causal Mechanisms.** Several papers explore systems with changing dynamics, either with an explicit notion of switching causal factors [Minka and Winn, 2008, Peters et al., 2016, Willig et al., 2025] or without [Chalupka et al., 2016, Seitzer et al., 2021, Liu et al., 2023]. The direct MCMs proposed here distinguish themselves from these by allowing for the explicit and predictable control of such qualitative changes from within the SCM, similar to the 'mechanized SCM' of Kenton et al. [2023] which aimed to model the influence of agentic agents over structural equations.

**Algorithmic Recourse and Fairness.** Our judicial decision-making example (Sec. 6.2) highlights the importance of reflecting on system dynamics in order to remedy situations that lead to unfair outcomes. This relates to work on fairness [Kusner et al., 2017, Zhang and Bareinboim, 2018, Von Kügelgen et al., 2022, Plecko and Bareinboim, 2023, 2024], treatment-confounder feedback in causal reinforcement learning [Bareinboim et al., 2015, Lu et al., 2018, Buesing et al., 2018, Zhang, 2020, Weichwald et al., 2022] and algorithmic recourse [Zhang and Bareinboim, 2022, Karimi et al., 2021, Von Kügelgen et al., 2022]. Our contribution to this space is providing a meta-causal framework for agents to not only optimize their actions with respect to some external metric, but be able to reflect and reason based on the meta-causal identification of edge activations.

# 8 Conclusion

This work addresses the critical shortcomings of classical causal analysis, which may rely on static assumptions, particularly in environments where interventions may alter the underlying causal dynamics of the system. We introduced a specialized class of meta-causal models (MCMs) designed to explicitly model and predict changes in evolving causal dynamics, formalized meta-causal variables –variables that govern the system's meta-causal state and are observable from within the SCM– and established conditions for meta-causal predictability. We presented a meta-causal analysis method, which, in contrast to prior work, also considers the intermediate meta-causal state dynamics which cause the eventually observed outcomes.

**Broader Impact.** We demonstrated the value of meta-causal decision-making through examples in medical treatment (Sec. 6.1) and judicial settings (Sec. 6.2). Beyond these cases, meta-causal analysis helps address the risks of ignoring system dynamics, enabling more sustainable and fair outcomes. The examples show how MCMs anticipate and adapt to shifting causal relationships, moving beyond static analyses.

**Limitations and Future Work.** The framework for meta-causal predictability, particularly the concept of Direct MCMs, relies on the strong assumption of full observability of meta-causal variables and the preservation of all information about transition factors via the abstraction functions. While concerns about the assertiveness of causal abstractions are common to the general field of causal representation learning [Rubenstein et al., 2017, Schölkopf et al., 2021, Kekić et al., 2023, Talon et al., 2024], complete observability may not be feasible in many real-world scenarios, and the factors driving meta-causal shifts might be latent or only partially captured. Further work on relaxing theoretical assumptions and empirical validation in a wider range of real-world settings may be necessary to achieve robustness of the proposed framework. Although the presented applications demonstrate the usefulness of MCM-based strategies in specific examples within confined simulated environments, transferring the proposed concepts to general meta-aware agents may be a promising avenue to explore in the future.

## Acknowledgments and Disclosure of Funding

The authors acknowledge the support of the German Science Foundation (DFG) research grant "Tractable Neuro-Causal Models" (KE 1686/8-1). This work was funded by the European Union (Grant Agreement no. 101120763 - TANGO). Views and opinions expressed are however those of the author(s) only and do not necessarily reflect those of the European Union or the European Health and Digital Executive Agency (HaDEA). Neither the European Union nor the granting authority can be held responsible for them. This work has benefited from the early stages of the fundings by the German Research Foundation (DFG) under Germany's Excellence Strategy— "Reasonable AI" (EXC-3057) and "The Adaptive Mind" (EXC-3066); funding will begin in 2026. The Eindhoven University of Technology authors received support from their Department of Mathematics and Computer Science and the Eindhoven Artificial Intelligence Systems Institute.

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

# When Causal Dynamics Matter: Adapting Causal Strategies through Meta-Aware Interventions (Appendix)

The appendix is structured as follows. App. A describes the implications of a non-bijectiveness on the prediction of future meta-causal states. App. B provides a description of the particular identification function used in the examples of this paper. App. C gives details on the SCM, starting conditions and simulation of the Medicating Flu Example. App. D details the SCM for the Judicial Decision-Making example. Similarly, App. E and App. F provide fully worked examples of applying meta-causal analysis to the two example scenarios of the main paper.

## A    Implications of a Non-Bijective Causal Abstraction

Works on causal abstraction often require the abstraction function $\varphi$ to be surjective Rubenstein et al. [2017], Beckers and Halpern [2019], Willig et al. [2023]. The more information about the initial process state is marginalized by $\varphi$, the less information about the original underlying process state can be recovered again from observations of the causal variables $\mathbf{x}$. A non-unique invertability of $\varphi$ does however not restrict the expressibility of MCM, but generalizes their prediction from a unique process state s for a given variable configuration $\mathbf{x}$ to the prediction of a distribution of possible states $\mathcal{P}(\mathrm{s})$ that all could have lead to the observed $\mathbf{x}$; $\varphi^{-1} : \boldsymbol{\mathcal{X}} \to \mathcal{P}(\mathrm{s})$. Unobserved or exogenous variables can equally be modeled, as variables which are marginalized by the abstraction function, thus being unavailable and inducing uncertainty when inferring the underlying system state via $\varphi^{-1}$.

Similar, to how non-determinism of the state transitions changes the output of the transition function to a probability distribution $\delta^{\boldsymbol{\mathcal{X}}} : \boldsymbol{\mathcal{X}} \to \mathcal{P}(\mathcal{T}^{N \times N})$ instead of predicting a unique $\mathcal{T}^{N \times N}$ (c.f. Sec. 3.1), a non-uniquely invertible $\varphi$ induced uncertainty over the function's input, such that $\delta^{\boldsymbol{\mathcal{X}}}$ changes from a deterministic input $\delta^{\boldsymbol{\mathcal{X}}} : \boldsymbol{\mathcal{X}} \to \mathcal{P}(\mathcal{T}^{N \times N})$ to $\delta^{\boldsymbol{\mathcal{X}}} : \mathcal{P}(\boldsymbol{\mathcal{X}}) \to \mathcal{P}(\mathcal{T}^{N \times N})$.

Within the proof of Thm.3.1 structural equations are given as $\mathbf{F} := \varphi \circ \sigma \circ \varphi^{-1}$. For the LMCD algorithm (Alg. 1) this means that an observed data point $\mathbf{x}^i$ can no longer be advanced deterministically to it next state $\mathbf{x}^{i,t+1}$, as the final $\varphi^{-1}$ component that $\mathbf{F}$ is composed of, now outputs a distribution of possible structural equations $\mathbf{F}_{T_{s,ij}}$ at a particular MCS $T_s$, according to some previously recorded probability distribution, $\mathbf{F}_{ij} \sim \mathcal{P}(\mathbf{F}_{T_{s,ij}})$. For advancing a particular data sample in line 3 of the LMCD algorithm, either a particular instantiation of structural equations $\mathbf{F}' \sim \mathbf{F}$ is (possibly repeatedly) sampled to approximate effects of the uncertain transition probabilities, or in case of a finite number of possible structural equations the sample is pushed-forward through $\mathbf{F}$ and the algorithm is continued with the resulting distribution.

## B    Choice of Identification Function in the Examples

The examples presented in this paper use an identification function that determines the presence of an edge by considering whether or not a non-zero gradient is present between any parent and child pair connected via a structural equation. The function therefore only emits the types 'effect exists' (1) or 'no effect' (0). For all other variable pairs which are not connected in the causal graph the 'no effect' type is always emitted. For any two continuous variables $X_i, X_j \in \mathbf{X}$ connected via a direct causal edge, $X_j \to X_i$, we use the 'contextual independency' function, defined as:

$$\mathrm{CIId}(X_i, X_j, \mathbf{x}) := (\frac{dX_i}{dX_j}(\mathbf{x}) \neq 0) \tag{6}$$

The function is true if the gradient of $X_j$ onto $X_i$ is non-zero, which is equal to $X_i$ being contextually dependent on $X_j$, given the subset of parent values of $X_i$ for the current variable configuration $\mathbf{x}$. Otherwise, it is false, indicating that no current causal influence of $X_j$ on $X_i$ is present. Furthermore, a corresponding discrete version can be defined as:

$$\mathrm{CIID}^{\mathrm{disc}}(X_i, X_j, \mathbf{x}) := (\exists x_j' \in X_j . (f_i(\mathbf{x}|_{\mathbf{X} \setminus X_j} \cup x_j') \neq f_i(\mathbf{x}))) \tag{7}$$

where $\mathbf{x}|_{\mathbf{X} \setminus X_j}$ are the values of $\mathbf{x}$ without $x_j$.

## C   Details on the Medicating Flu Example

This section provides further details on the simulations discussed Sec. 6.1 and shown in Fig. 2. All discussed results are averaged values from 1,000 independent roll-outs of the described system. Noise for every roll-out is individually seeded and sampled independently. Code to reproduce the experiments can be found in the supplementary material. For simulations the following exact structural equations where evaluated for 10 discrete time series:

$$\text{Exposure}_t := 0.8 \cdot \text{Binomial}(n = 10, p = 0.5) \tag{8}$$

$$\text{ImmuneStrength}_t := \text{ImmuneStrength}_{t-1} + \text{Normal}(t; \mu = 5, \sigma = 2) - \beta_k \text{Medication}_{t-1} \tag{9}$$

$$\text{Fever}_t := \max(\text{Exposure}_t - \text{ImmuneStrength}_t,\ 0) \tag{10}$$

$$\text{Medication}_t := \begin{cases} 1 & \textbf{if } \text{Fever}_t > 0.5 \\ 0 & \textbf{otherwise} \end{cases} \tag{11}$$

$$\text{BodyTemperature}\Delta_t := \text{Fever}_t \cdot ((1 - \alpha_k) \cdot \text{Medication}_t) \tag{12}$$

The starting values for all variables, except ImmuneStrength, before advancing to the first timestep are set to $0.0$ (and $2.0$ for ImmuneStrength). Timesteps start a zero and are evaluated over 10 time steps, with each timestep representing a two year span and the final step $t = 9$ ending at age 18. All other values follow from these initial values with conjunction with randomly sampled exposure levels at every timesteps.

**Compute Resources.** Simulations and visualizations run in under 5 seconds on PC with a AMD Ryzen Threadripper 1900X 8-Core Processor and 32GB of RAM.

## D   Details on the Judicial Decision-Making Example

This section contains the accompanying set of structural equations for the visualization in Fig. 1, Sec. 6.2 and Fig. 4:

$$\text{CasePool}_t := \text{CasePool}_{t-1} \setminus \{\text{CasePool}_{t-1}[\text{Schedule}_{t-1}]\} \tag{13}$$

$$\text{Schedule}_t^{\text{initial}} := 0 \tag{14}$$

$$\text{Schedule}_t^{\text{adapted}} := \operatorname{argmax}(\text{CasePool}_t) \tag{15}$$

$$\text{CaseComplexity}_t := \text{CasePool}_t[\text{Schedule}_t] \tag{16}$$

$$\text{Fatigue}_t := \text{Fatigue}_{t-1} + 0.5 \tag{17}$$

$$\text{DecisionBias}_t := \max(\text{Fatigue}_t + \text{CaseComplexity}_t - 5, 0) \tag{18}$$

Time $t$ is evaluated from $[0..5]$. All values, except for the case pool, are assigned value $0$ before the first timestep. The case pool at timestep 0 is set as a random permutation over a fixed set of cases:

$$\text{CasePool}_0 := \operatorname{permute}([3, 2, 4, 1, 3, 4]) \tag{19}$$

Fig. 1 in the main paper, shows the roll-out of the scenario with unpermuted values.

## E   Worked example: Medicating Flu

In this section we provide a fully worked example for applying MCA for the medicating flu scenario of Sec. 6.1. Code for reproducing all examples is provided at: `https://github.com/MoritzWillig/metaCausalDynamics`. We start off with the SCM described in App. C. For identifying the activation of edges we utilize the CIId identification function described in App. B. As a result, we identify effects via the following influence functions:

$$\text{ExposureInf}_t := \{\} \tag{20}$$

$$\text{ImmuneStrengthInf}_t := \left\{ \begin{array}{c} \text{Time}_t : 1 \\ \text{Imm.Str.}_{t-1} : 1 \\ \text{Medication}_{t-1} : \text{Medication}_{t-1} \neq 0 \end{array} \right\} \tag{21}$$

$$\text{FeverInf}_t := \left\{ \begin{array}{c} \text{Exposure}_t : \text{Exposure}_t > \text{Imm.Str.}_t \\ \text{ImmuneStr}_t : (\text{Exposure}_t > 0) \wedge (\text{Imm.Str.}_t > 0) \end{array} \right\} \tag{22}$$

$$\text{MedicationInf}_t := \{ \qquad \text{Fever}_t : \text{Fever}_t > 0.5 \qquad \} \tag{23}$$

$$\text{BodyTemperature}\Delta\text{Inf}_t := \left\{ \begin{array}{c} \text{Fever}_t : \text{Fever}_t > 0 \\ \text{Medication}_t : (\text{Fever}_t > 0) \wedge (\text{Medication}_t > 0) \end{array} \right\} \tag{24}$$

The functions determine the influence of the respective parents onto the variables. The entry $\text{ImmuneStrengthInf}_t := \{\text{Imm.Str.}_{t-1} : 1\}$, for example, indicates that the edge from immune strength in the last timestep onto immune strength in the current timestep is always active. All remaining edges not identified by any of the above influence functions are identified as $0$.

**Defining the MCM.** The example models a direct MCM (Def. 4) with $\varphi = \text{Id}$, such that the mediation process directly becomes the SCM. With variables $\mathbf{X} = \{\text{Time, Exposure, ImmuneStrength, Fever,}$ Medication, BodyTemperature$\Delta\}$ and structural equations as given in App. C the mediating process is: $\mathcal{E} = (\mathbf{X}, \mathbf{F}_{\text{Eqs. } 8-12})$. The Meta-Causal frame is then defined as

$$\mathcal{F} = (\mathcal{E}, \mathbf{X}, \tau_{\text{Eqs. } 20-24}, \mathcal{I}) \tag{25}$$

and $\mathcal{I}(s, X_i, X_j) \mapsto t := \tau_{ij}(\varphi(s), \varphi \circ \sigma) = \tau_{ij}(\mathbf{x}, \mathbf{F})$ according to Def. 1. The identification function identifies the pure presence or absence of edges $T_{s,ij} \in \{0, 1\}$, such that a meta-causal state is given as

$$\mathrm{T} \in \mathcal{T}^{2|\mathbf{X}| \times 2|\mathbf{X}|} = \{0, 1\}^{12 \times 12} \tag{26}$$

Note that the MCS has double the entries as there are variables in a single timestep, since effects are identified between the current, but also to variables of the previous timestep. Finally, we define the Meta-Causal Model:

$$\mathcal{A} = (\mathcal{T}^{2|\mathbf{X}| \times 2|\mathbf{X}|}, \mathbf{X}, \sigma) = (\{0, 1\}^{12 \times 12}, \mathbf{X}, \mathbf{F}) \tag{27}$$

### E.1 Meta-Causal Analysis.

We apply the LMCD algorithm (Alg. 1) under the previously defined MCM. The same 1000 roll-outs for both medications A and B as in Sec. 6.1 are sampled from the SCM. Environment states at the final timestep ($t = 9$) are then selected for further analysis, $\mathbf{x}^{\mathbf{I}} = (\mathbf{x}^{i,t=9})_{i=1}^{1000}$.

Identifying all meta-causal states –the unique sets of active edges in a sample according to the above influence functions– of all samples $\mathbf{x}^i \in \mathbf{x}^{\mathbf{I}}$ yields 10 unique MCS for drug A and 8 unique MCS for drug B. For this analysis, we focus on the Fever $\rightarrow$ Temperature and Medication $\rightarrow$ Temperature edges within the meta-causal states of $T^i := \mathcal{I}(x^i)$ and $T^{i,t+1} := \mathcal{I}(x^{i,t+1})$. This is realized via the indexing function $l$ within the LMCD algorithm.

$$l : T \mapsto [T_{\text{Fever,Temperature}}, T_{\text{Medication,Temperature}}] \tag{28}$$

The resulting vector can take the four distinct states $[0, 0]$ "no fever (no treatment)", $[1, 0]$ "mild fever (no treatment)", $[1, 1]$ "fever with medication" and $[0, 1]$ "treatment without fever". As there is never a treatment without fever in the data, the LMCD algorithm identifies the following set of unique MCS:

$$U = \{[0, 0], [1, 0], [1, 1]\} \tag{29}$$

Counting the occurrence of individual meta-causal states at timestep $t = 9$ ("age 18") obtains the following MCS counts:

$$C_A^{t=9} = \begin{bmatrix} 10 \\ 9 \\ 981 \end{bmatrix}; \quad C_B^{t=9} = \begin{bmatrix} 437 \\ 104 \\ 459 \end{bmatrix} \tag{30}$$

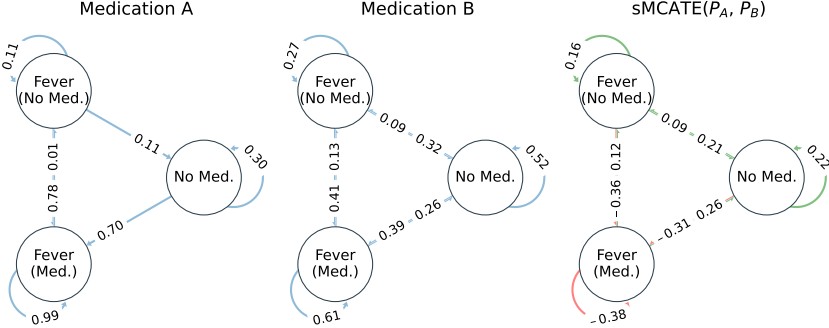

Figure 5: **Medicating Flu Transitions.** Transition probabilities between different meta-causal states, visualized as Markov processes for medication A, medication B and the resulting sMCATE($P_A, P_B$) of the Medicating Flu example in Sec. 6.1.

After advancing the individual samples in time $\mathbf{x}^{i,t+1} := \mathbf{F}((\mathbf{x}^i\,|\mathbf{v}) \cup (\mathbf{u}^{t+1} \sim P_\mathbf{U}))$, and computing its MCS $\mathrm{T}^{i,t+1} := \mathcal{I}(\mathbf{x}^{i,t+1})$, we obtain the following absolute transition counts:

$$C_A = \begin{bmatrix} 3 & 0 & 7 \\ 1 & 1 & 7 \\ 0 & 6 & 975 \end{bmatrix} ; \quad C_B = \begin{bmatrix} 227 & 40 & 170 \\ 33 & 28 & 43 \\ 120 & 58 & 281 \end{bmatrix} \tag{31}$$

The per-row normalized transition matrices then model the transition probabilities at time $t$:

$$P_A = \begin{bmatrix} 30.00\% & 0.00\% & 70.00\% \\ 11.11\% & 11.11\% & 77.78\% \\ 0.00\% & 0.61\% & 99.39\% \end{bmatrix} \tag{32}$$

$$P_B = \begin{bmatrix} 51.95\% & 9.15\% & 38.90\% \\ 31.73\% & 26.92\% & 41.35\% \\ 26.14\% & 12.64\% & 61.22\% \end{bmatrix} \tag{33}$$

Optionally, the continuous-time rate matrices are computed as $Q := e^{P-I}$. (For this discrete time scenario $P$ might be sufficient):

$$Q_A = \begin{bmatrix} 0.496 & 0.001 & 0.502 \\ 0.050 & 0.412 & 0.537 \\ 0.000 & 0.004 & 0.995 \end{bmatrix} \tag{34}$$

$$Q_B = \begin{bmatrix} 0.662 & 0.066 & 0.271 \\ 0.211 & 0.506 & 0.282 \\ 0.186 & 0.082 & 0.731 \end{bmatrix} \tag{35}$$

Finally, the specific Meta-Causal ATE expresses the change in transition probabilities effect between both medications $A$ and $B$:

$$\text{sMCATE}(P_A, P_B) = P_B - P_A = \begin{bmatrix} 21.94\% & 9.15\% & -31.09\% \\ 20.61\% & 15.81\% & -36.43\% \\ 26.14\% & 12.02\% & -38.16\% \end{bmatrix} \tag{36}$$

The obtained transition probabilities $P_A, P_B$ and their difference, $sMCATE_{disc}(P_A, P_B)$, can be visualized as Markov processes. The corresponding graphs are shown in Fig. 5.

**Interpretation.** From the $C$ matrices we see that the third state, indicating fever induced medication, is almost always active for drug A (98.1%), while it is only given in 45.9% of the cases for drug B. Considering the meta-causal transition matrices $P_A, P_B$, we do not only find that the MCS of fever induced medication is increased in A, but also that all other states eventually flow into this state (with probabilities of 70.0% and 77.78%). Drug B therefore features more favorable dynamics. Here, patients with a fever induced medication are still rather likely to to stay in that state. However, (moderately) healthy patients only transition into this state with moderate probabilities of 38.9% and 41.35% and are similarly able to leave again from it. These differences in dynamics are also visible from the sMCATE, which shows a sharp decrease in transition probabilities into the third state, while the healthy states observes a strong increase in incoming transition probabilities.

# F  Worked example: Judicial Decision Making

In this section we provide a fully worked example for applying MCA for the judicial decision making scenario of Sec. 6.2. Code for reproducing all examples is provided at: `https://github.com/MoritzWillig/metaCausalDynamics`. We start off with the SCM described in App. D. For identifying the activation of edges we utilize the CIId identification function, as before, described in App. B. As a result, we identify effects via the following influence functions:

$$\text{CasePoolInf}_t := \begin{cases} \text{CasePool}_{t-1}\colon 1 \\ \text{Schedule}_{t-1}\colon 1 \end{cases} \tag{37}$$

$$\text{ScheduleInf}_t^{\text{initial}} := \{\} \tag{38}$$

$$\text{ScheduleInf}_t^{\text{adapted}} := \{\text{CasePool}_t\colon 1\} \tag{39}$$

$$\text{CaseComplexityInf}_t := \begin{cases} \text{CasePool}_t\colon 1 \\ \text{Schedule}_t\colon 1 \end{cases} \tag{40}$$

$$\text{FatigueInf}_t := \{\text{Fatigue}_{t-1}\colon 1\} \tag{41}$$

$$\text{DecisionBiasInf}_t := \begin{cases} \text{Fatigue}_t\colon (\text{Fatigue}_t + \text{CaseComp.}_t \geq 5) \wedge (\text{Fatigue}_t \neq 0) \\ \text{CaseComp.}_t\colon (\text{Fatigue}_t + \text{CaseComp.}_t \geq 5) \wedge (\text{CaseComp.}_t \neq 0) \end{cases} \tag{42}$$

The functions determine the influence of the respective parents onto the variables. The entry $\text{CasePoolInf}_t := \{\text{CasePool}_{t-1} : 1\}$, for example, indicates that the edge from the case pool in the last timestep onto the case pool in the current timestep is always active. All remaining edges not identified by any of the above influence functions are identified as $0$.

**Defining the MCM.** The example models a direct MCM (Def. 4) with $\varphi = \text{Id}$, such that the mediation process directly becomes the SCM. With variables $\mathbf{X} = \{\text{CasePool}, \text{Schedule}, \text{CaseComplexity}, \text{Fatigue}, \text{DecisionBias}\}$ and structural equations as given in App. C the mediating process is: $\mathcal{E} = (\mathbf{X}, \mathbf{F}_{\text{Eqs. } 13-18})$. The Meta-Causal frame is then defined as

$$\mathcal{F} = (\mathcal{E}, \mathbf{X}, \tau_{\text{Eqs. } 37-42}, \mathcal{I}) \tag{43}$$

and $\mathcal{I}(s, X_i, X_j) \mapsto t := \tau_{ij}(\varphi(s), \varphi \circ \sigma) = \tau_{ij}(\mathbf{x}, \mathbf{F})$ according to Def. 1. The identification function identifies the pure presence or absence of edges $T_{s,ij} \in \{0, 1\}$, such that a meta-causal state is given as

$$\mathbf{T} \in \mathcal{T}^{2|\mathbf{X}| \times 2|\mathbf{X}|} = \{0, 1\}^{10 \times 10} \tag{44}$$

Note that the MCS has double the entries as there are variables in a single timestep, since effects are identified between the current, but also to variables of the previous timestep. Finally, we define the Meta-Causal Model:

$$\mathcal{A} = (\mathcal{T}^{2|\mathbf{X}| \times 2|\mathbf{X}|}, \mathbf{X}, \sigma) = (\{0, 1\}^{10 \times 10}, \mathbf{X}, \mathbf{F}) \tag{45}$$

## F.1  Meta-Causal Analysis.

We apply the LMCD algorithm (Alg. 1) under the previously defined MCM. Roll-outs for all 720 initial case pool permutations are sampled from the SCM for both, the eager and reflected, policy. States transition across all timesteps and roll-out are considered for further analysis, $\mathbf{x}^{\mathbf{I}} = (\mathbf{x}^i)_{i=1}^{2880}$.

Identifying all meta-causal states –the unique sets of active edges in a sample according to the above influence functions–, of all samples $\mathbf{x}^i \in \mathbf{x}^{\mathbf{I}}$, via $T^i := \mathcal{I}(x^i)$ and $T^{i,t+1} := \mathcal{I}(x^{i,t+1})$, observes 2 unique MCS for the eager policy. Either the 'unbiased' state with no influence of fatigue or case complexity on the decision bias, or the 'biased' state where both parents exert influence on the decision bias variable. The reflected policy always stays within the 'unbiased' MCS. We write $[0]$ and $[1]$ to identify the unbiased and biased MCS, instead of writing down the whole $\{0, 1\}^{10 \times 10}$ type matrices in the following:

$$U = \{[0], [1]\} \tag{46}$$

Counting the occurrence of individual meta-causal states, advancing the individual samples in time $\mathbf{x}^{i,t+1} := \mathbf{F}((\mathbf{x}^i|_{\mathbf{v}}) \cup (\mathbf{u}^{t+1} \sim P_{\mathbf{U}}))$, and computing their MCS $T^{i,t+1} := \mathcal{I}(\mathbf{x}^{i,t+1})$ obtains the following absolute transition counts:

$$C_{\text{eager}} = \begin{bmatrix} 504 & 936 \\ 576 & 864 \end{bmatrix}; \quad C_{\text{reflected}} = \begin{bmatrix} 0 & 0 \\ 0 & 2000 \end{bmatrix} \tag{47}$$

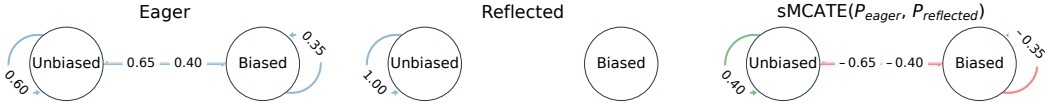

Figure 6: **Judicial Decision Making Transitions.** Transition probabilities between different meta-causal states, visualized as Markov processes for the eager and reflected strategies of the Judicial Decision Making example in Sec. 6.2.

The per-row normalized transition matrices then model the state changes probabilities:

$$P_{\text{eager}} = \begin{bmatrix} 35.0\% & 65.0\% \\ 40.0\% & 60.0\% \end{bmatrix} \tag{48}$$

$$P_{\text{reflected}} = \begin{bmatrix} 0.0\% & 0.0\% \\ 0.0\% & 100.0\% \end{bmatrix} \tag{49}$$

Optionally, the continuous-time rate matrices are computed as $Q := e^{P-I}$. (For this discrete time scenario $P$ might be sufficient):

$$Q_{\text{eager}} = \begin{bmatrix} 0.597 & 0.402 \\ 0.247 & 0.752 \end{bmatrix} \tag{50}$$

$$Q_{\text{reflected}} = \begin{bmatrix} 0.367 & 0.0 \\ 0.0 & 1.0 \end{bmatrix} \tag{51}$$

Finally, the specific Meta-Causal ATE expresses the change in transition probabilities effect between the eager and reflected policy:

$$\text{sMCATE}(P_{\text{eager}}, P_{\text{reflected}}) = P_{\text{reflected}} - P_{\text{eager}} = \begin{bmatrix} -35.0\% & -65.0\% \\ -40.0\% & 40.0\% \end{bmatrix} \tag{52}$$

The obtained transition probabilities $P_{\text{eager}}, P_{\text{reflected}}$ and their difference, $sMCATE_{disc}(P_{\text{eager}}, P_{\text{reflected}})$, can be visualized as Markov processes. The corresponding graphs are shown in Fig. 6.

**Interpretation.** The absolute MCS state counts in $C_{\text{eager}}$ indicate a $40.0\%$ chance of remaining with a biased decision in the eager policy case. The transition dynamics $P_{\text{eager}}$ and $P_{\text{reflected}}$ furthermore indicate a moderate chance ($40\%$ and $65\%$) of transitioning between the two MCS at every timestep. Conversely, the reflected policy remains solely within the unbiased state, fully eliminating all other transitions. This is similarly reflected in the sMCATE, as offsets in the transition probabilities –except for the unbiased $\rightarrow$ unbiased self-cycle– fully counteract all other transition probabilities of $P_{\text{eager}}$.

