# OpenReview forum: "When Causal Dynamics Matter: Adapting Causal Strategies through Meta-Aware Interventions"
_NeurIPS.cc/2025/Conference — NeurIPS 2025 poster_

### Official Review · Reviewer_jRAi · 2025-06-03

**Clarity:** 3
**Significance:** 3
**Originality:** 3
**Rating:** 4
**Confidence:** 3

**Summary:**

This paper studies the modeling and prediction of intervened outcomes using meta-causal models (MCM).

**Questions:**

See weakness.

**Ethical Concerns:**

["NO or VERY MINOR ethics concerns only"]

**Final Justification:**

I will keep my positive score.

**Limitations:**

yes

**Quality:**

3

**Strengths And Weaknesses:**

**Strength**

This paper provides very interesting for dynamic causal inference.

**Weakness**

This paper should clearly articulate the underlying assumptions used for causal inference, as well as the limitations of the proposed approach.
1. My main concern lies in the bijectiveness of \varphi. How is this assumption justified within the context of meta-causal models (MCM)? Could you provide examples of cases in which the assumption is satisfied and cases in which it is violated? This is crucial for assessing the reliability of the results.
2. The identification of the average treatment effect (ATE) typically requires an exogeneity or ignorability assumption. Are such assumptions not required in your paper?
I believe that such assumptions are necessary, and that the meta-causal framework (MCF) exclude the problem of confounding.
3. A substantial body of work by James Robins focuses on causal inference in time series and longitudinal settings. A discussion on the differences among his studies is necessary to clarify their respective contributions and methodological distinctions.
4. Additionally, similar studies have been conducted under the framework of “longitudinal data” in the field of statistics.

---

> ### Author Rebuttal · Authors · 2025-07-31
>
> Dear Reviewer,
> thank you for your feedback on our paper. We will try to address your mentioned weaknesses/questions in the following:
>
> 1) **W1 Bijectiveness of \\varphi:** Bijectiveness of \\varphi does not restrict the expressibility of MCM, but rather decides which meta-causal states and future transitions can be predicted from (endogenous) observations of the SCM's variables. This observability is 1) necessary to estimate the causal type of an edges to determine the current meta-causal type (structural equation to MCS) and therefore the transition probabilities (c.f. steps 2.1 and 2.2 of the Linearized-Meta-Causal-Dynamics algorithm in our reply to reviewer xvfw). This that this direction does not exclude the presence of exogenous factors, (consider exogenous exposure levels in the Flu example), but determined the finegrainedness with which meta-causal types can be identified. 2) In the reverse direction (MCS to structural equation) consider that a particular meta-causal state $T_{s+1,ij}$ is sampled in step 2.1 of the algorithm. However, without being able to invert \\varphi the particular equations for the process values $s$ can not be determined, such that these values and their subsequently emitted influence on edge type can no longer be determined.
>
>    We take your comment as a suggestion to relax the requirement of \\varphi as to only be invertible. We therefore drop the uniqueness requirement. At the time of advancing the system state, this allows to sample an actual structural equation $F\_{ij}$ from a set of candidate equations $\\textbf{F}\_{T\_{s,ij}}$ that have  been observed earlier for a particular edge type $T_{s,ij}$, according to some previously recorded probability distribution, $F_{ij} \\sim P_{\\textbf{F}\_{T\_{s,ij}}}$. This introduces uncertainty at the level of configuration of the SCM in terms of structural equations and allows for a more broader area of application. We incorporated the discussed changes in the definition and follow up texts, and will discuss their implications in an additional section in the appendix.
>
> 2) **W2 Identification of the average treatment effect.** Since building meta-causal models upon the Pearlian causal framework ATE computations can be computed over the causal graph. Within the Pearlian framework the do-calculus given the sufficient and necessary conditions for such causal effect estimations under which condition unbiased (e.g. unconfounded estimators can be obtained). The mentioned exogeneity and ignorability can be directly derived from the do-calculus, and we assumed them to hold implicitly. However, we agree that these implicit assumption might be not obvious and potentially confusing. We added a corresponding note on the made assumption and potential use of do-calculus before introducing ATE calculations. Thanks for pointing this out!
>
> 3) **W3/W4 Related Works by James Robins / Longitudinal data in Statistics.** Thanks for pointing us to these works and general methods on longitudinal data. We agree that these methods in longitudinal statistics are highly related to our ideas. Although a strong overlap in both areas of longitudinal and meta-causal analyses (MCA) exists, both approaches are trying to answer different questions in their respect. Nonetheless, we will supplement our related work section with a summary of the following references, and provide a more detailed discussion in a new appendix section:
>    **Robins' works and general Longitudinal Statistics** cover influential works on the g-formula [1,2] that aims to estimate exposure effects in the presence of time-varying confounders. Most similar to out work might be the paper of Robins and James [3] in which similarly leverages marginal structural models to be used for causal effect estimation under time-dependent confounding. Similarly, the works of Kung-Yee and Zegler [4] and Laird and Ware [5] utilize linearized models and random-effect sampling to approximate the effects of long term dynamical shifts in the resulting distributions and corresponding ATE.
>    In **contrast to prior works**, our paper focuses on performing Meta-Causal Analysis (MCA). MCA is not a separate set of analyses methods, but designed to be used in combination and to supplement existing methods. More specifically, MCA considers the transition dynamics between different meta-causal states (/different structural configurations of a system) at a particular points in time. Our newly proposed algorithm, linearizes transition dynamics after the possible unrolling of a time series model to determine the transition probabilities between different meta-causal states.  In this regard please note the newly proposed algorithm on performing MCA in our response to reviewer xvfw, as well as the resulting worked out examples in response to reviewer CdGz. Eventually these state transitions translate to switching structural equations, that, in return, induce particlar variable distributions, which can finally be captured via the suggested parametric g-formula, dynamic treatment regime methods and similar statistical analyses. (This is what we had computed so far in our paper). However, MCA is not only interested in the resulting actual effects, but also how such effects come to be with the help of graphical causal models, e.g. whether particular mechanisms are active at particular points in time, which might be either desired or undesired. To better highlight possible questions to be answered by MCA, we partially repeat our answer to reviewer CdGz: Natural question to answer with MCA would be "How likely is a system to adapt a desired state?", "How stable is a desired state?" or "Which paths (in terms of MCS sequences) are available to reach a particular state?" (and "Are these paths admissible in terms of intermediate system behavior? E.g. are they safe to navigate under safety or ethics standpoints?").
>    To the best of our knowledge, the previous mentioned questions have not yet been answered in a unified framework by previous approaches. The explicit graphical modeling of (meta-causal) state transitions in causal systems is not covered in the previously suggested fields of work and constitutes the main novelty of our work. Kindly note that we gave a similar answer to reviewer CdGz which pointed us in the same direction.
>
>
> The above answers outline the role of meta-causal analyses in relation to existing methods in longitudinal statistics and dynamic treatment regimes, and separate them from the novel contributions of our paper in terms of MCA. We elaborated on the role of varphi and the relation of ATE and the graphical causal models used in our work. Please let us know in case me missed any key aspects or papers, or you want to discuss any of the given points further.
>
> Best regards,
> the authors
>
>
> [1] Robins, James. "A new approach to causal inference in mortality studies with a sustained exposure period—application to control of the healthy worker survivor effect." Mathematical modelling 7.9-12 (1986): 1393-1512.
> [2] Robins, James M. "Causal inference from complex longitudinal data." Latent variable modeling and applications to causality. New York, NY: Springer New York, 1997. 69-117.
> [3] Robins, James M., Miguel Angel Hernan, and Babette Brumback. "Marginal structural models and causal inference in epidemiology." Epidemiology 11.5 (2000): 550-560.
> [4] Liang, Kung-Yee, and Scott L. Zeger. "Longitudinal data analysis using generalized linear models." Biometrika 73.1 (1986): 13-22.
> [5] Laird, Nan M., and James H. Ware. "Random-effects models for longitudinal data." Biometrics (1982): 963-974.

---

> > ### Comment · Reviewer_jRAi · 2025-08-03
> >
> > Thank you for your comment. Keep my score.
> > How will you discuss this assumption within the SCM framework?

---

> ### Author Response · Authors · 2025-08-04
>
> Dear reviewer,
> thank you for your reply. We are not quite sure if you implied any particular assumption to be discussed within the SCM framework, but we will naturally implement all proposed changes in the paper. As our whole work is build upon the Pearlian SCM framework, we will frame all proposed changes within that. Specifically, we will make the following modifications:
>
> * **W1)** The relaxation to invertible varphi will be implemented directly within the definitions. We will briefly mention implications in the main text and provide the slightly longer discussion, that we gave in the above reply, on the resulting implications and special case of bijectiveness in the appendix.
> * **W2)** We will briefly clarify causal effect identification of ATE beyond the bivariate case, explicitly hinting to the full use of do-calculus.
> * **W3)** We will use some of the extra space of the camera-ready version to discuss related work in longitudinal statistics. Generally, we will change the existing related work section to be more constrastive as discussed/proposed by you.
> * **MCA)** We will reserve the majority of the additional space to add an new subsection on meta-causal analysis and the previously discussed perspectives, together with the newly proposed algorithm, before moving on to the experiments. We will provide the presented worked out examples in the appendix and discuss briefly qualitatively in the main text at the end of the corresponding experiment.
>
> Please let us know if you would like us to go into more detail on any of the points above.

---

### Official Review · Reviewer_D4bW · 2025-06-20

**Clarity:** 1
**Significance:** 2
**Originality:** 2
**Rating:** 4
**Confidence:** 2

**Summary:**

Modelling a dynamic environment with traditional causal modelling (SCMs) assumes that the causal structure remains the same over time. In other words, causal dynamics are static and not affected by decisions under standard SCMs. Meta Causal Models (MCMs) alleviate the former concern by allowing for the causal dynamics to evolve over time, but do not resolve the latter. This paper introduces a new class of MCMs rich enough to characterize intervnentions and their long-term effects on the structure of a dynamic causal system.

**Questions:**

I was wondering if the dynamic environments from your paper and the effect of interventions on their underlying causal struture could be modelled by mechanised SCMs [1] where object level variables can directly affect mechanism level variables. Could you maybe share your on thoughts on that, and explain why the increased complexity of Direct MCMs is necessary?

[1] Discovering Agents, Kenton et.al., 2022

**Ethical Concerns:**

["NO or VERY MINOR ethics concerns only"]

**Final Justification:**

Clarity of the paper was improved with additional discussion and examples.

**Limitations:**

Adequately discussed

**Paper Formatting Concerns:**

All good

**Quality:**

3

**Strengths And Weaknesses:**

+ The problem being addressed is well explained and adequately motivated
+ The analysis of the proposed extension is detailed and deep
+ The applications section provides good insights about the proposed approach

- I found the paper quite heavy to read without familiarity with prior work. This is not necessarily bad, but it was difficult for me (and audience with background in more "traditional" causal inference) to follow the techical details and appreciate the importance of the contributions.
- The complexity of the proposed framework together with the paper's very limited empirical evaluation raises concerns about the pracicality of this approach. (See questions for an alternative)

---

> ### Author Rebuttal · Authors · 2025-07-31
>
> Dear Reviewer,
> thank you for your feedback on our paper. We will try to answer your mentioned weaknesses and questions in the following:
>
> * **W1: Heavy to read.** We agree that the paper is quite theoretical in its initial definitions. We tried to counter this fact, by providing two applied examples that show practical aspects and come with less formalism. Given your feedback, we will use some of the additional space of the camera ready version to contextualize definitions better and give interpretations for the particular modeling choices of direct MCM. (For example, see our reply to reviewer jRAi on the role and possible relaxation of the bijectiveness assumption of the varphi function.) Generally speaking, (direct) meta-causal models determine (or inversely infer) the functional/mechanistic relations of (or from) a system via abstracted function types. These types are then used to qualify the overall system states and transition between them, to qualitatively model their (meta-causal) dynamics. We will expand on these aspects, before starting with the formal definitions.
>
> * **W2: Practicality of the Approach.** MCMs are a rather new concept that is under current development and focuses on the qualitative state transitions. We are aware of the current rather theoretical type of papers, but hope to have presented two rather intuitive (and less theory heavy examples). In response to your and the other reviewers' remarks **we provide a Linearized-Meta-Causal-Dynamics algorithm (see our response to reviewer xvfw) and worked out examples (see our response to reviewer CdGz).** We will supplement the paper with these new methods and calculations to better highlight the actionability of our approach. Similarly to our reply to reviewer CdGz we want to elaborate further on the distinction between classical dynamical analyses and meta-causal analyses (MCA), in that are concerned in the transition of qualitative system dynamics rather than (and while intimately linked) the measurement of actual emitted effects via such changes. Natural questions to answer with MCA would be "Is a system able to obtain a desired state?", "How stable is a desired state?" or "Which paths (in terms of MCS sequences) are available to reach a particular state?" (and in this context "Are these paths admissible in terms of intermediate system behavior? E.g., are they admissible to navigate under general safety or ethics standpoints?").
>
> * **Q1: Dynamic Environments / Relation to Mechanized SCM.** While MCM eventually relate to a multitude of prior existing frameworks, we recognize that we did not consider the mentioned mechanized SCM as a rather closely related idea. As MCM initially come without explicit mechanism level variables by switching mechanism directly without the intermediate consideration of mechanistic variable, they might be more compact in representation at the expense of reduced interpretability.
>
>   While general MCM [allow for causal abstraction of the data generation process our direct MCM enforce explicit control of the endogenous variables over the structural equations and are, therefore, closer to mechanized SCM. By making the transition factors explicitly observable, we believe that both frameworks might eventually be transitioned into each other. Depending on the use case (e.g., modeling general system dynamics from observations versus modeling the influence of agentic agents on the system), one or the other representation might be better suited in certain situations. Thanks again for pointing us to this line of work. We will add the above discussion to the final paper.
>
>
> Thank you once again for your helpful feedback on our paper. We hope to have answered your questions to a satisfactory degree and are happy to elaborate further, in case any open points remain.
>
> Best regards,
> the authors

---

> > ### Author Response · Authors · 2025-08-04
> > **Any further concerns?**
> >
> > Dear reviewer,
> >
> > Since the discussion period will end in a couple of days, we would like to ask if there are any further questions from your side. We have replied in detail to all your original concerns and hope that we have alleviated your concens. We would be happy to discuss further if necessary.
> >
> > Regards,
> >
> > The authors

---

> > > ### Comment · Reviewer_D4bW · 2025-08-04
> > >
> > > I thank the authors for their detailed answer. I am covered wrt the mechanized SCMs connection. I also appreciate the new examples. The new algorithm I think is now too much of an addition to take into consideration during rebuttal. I will raise my score to 4 since my main concern was clarity issues, and this seems to have been improved by the authors.

---

### Official Review · Reviewer_xvfw · 2025-06-25

**Clarity:** 2
**Significance:** 2
**Originality:** 1
**Rating:** 2
**Confidence:** 3

**Summary:**

The paper addresses the theme of interventions that change the underlying system dynamic. In order to reason about evolving relationships, they introduce the formalism of meta-causal models and discuss a specific subclass of those with desirable properties. Furthermore, the authors discuss a few examples where a meta-causal lens is desirable.

**Questions:**

Barring the addition of more content (results) there is not a lot to do on the paper to improve it I would say. See above for suggestions on what could be added.

**Ethical Concerns:**

["NO or VERY MINOR ethics concerns only"]

**Final Justification:**

After reading the rebuttal and the other replies/reviews, I am still of the opinion that the original paper does not stand on its own, and that too much content needs to be added to make this a valuable contribution that passes the bar of NeurIPS. I see the authors have put a lot of effort in describing the revisions they intend to make, which is laudable, but I still have trouble envisioning what the paper will look like afterwards and some key additions--like the new algorithm--cannot be properly assessed during such a rebuttal. The way I see it, the bulk of the work needs to be present in the original paper, while here it looks like the main innovations (the new algorithm, the sMCATE) the motivation (what questions does this new framework address) and the convincing examples (showing how the old frameworks cannot do that they could do) have all been given in the rebuttal. In light of all this, I'll keep my score, but happy to discuss more with AC and other reviewers.

**Limitations:**

The societal impact and limitations have been addressed satisfactorily.

**Paper Formatting Concerns:**

Formatting is fine.

**Quality:**

1

**Strengths And Weaknesses:**

Clarity - The paper is written clearly and there seems to be enough attention on relevant literature.

Originality - The paper proposes a formalism that is novel as far as I know.

Significance - The topic of dynamical causal systems with feedback mechanisms is definitely a significant one.

Despite these strengths, I have some comments and concerns. The first one is that this paper is rather thin on contributions. Beside the definitions, there is only one theorem, whose proof is rather trivial. Even though the examples are interesting, described and contextualized well, they are not linked to the theory because the theoretical formalism is not used; it is therefore not demonstrated that the formalism proposed in the earlier section is needed or better than other options. Hence, I believe this may not be enough meat for a full paper at NeurIPS. To consider this for a full paper I would want to see some more results on the theory and a full formalisation of the examples, at least.

My second concern is that, after reading the examples, I do not understand why the new formalism is needed at all. These seem to be two cases in which we have an SCM unfolded over discrete time steps, and we are interested in the effect of interventions over long-term outcomes (frequency of fever episodes in the first case, and amount of biased decisions overall in the second case). It seems to me this can be modelled satisfactorily with the current SCM formalism by taking a (simple) dynamic treatment regime (DTR) perspective. That is, for example in the flu scenario, by considering interventions not at a single time point (do I give drug A or B at time t) but interventions that change the causal mechanism of the Medication variable uniformly at all time points: in the paper’s table 2, the strategies seem to be “always give drug A” and “always give drug B”. Equivalently, one can think of this as a bundle of hard interventions on all the Medication variables, one per time point, setting each to a constant function = A or =B respectively. Of course, one can also think of DTRs that are more complex (e.g. give drug A if fever is above threshold Z, etc). We can then check end-point outcomes or long-term outcomes depending on what we are interested in, to see which DTR is better. These are classical problems in epidemiology, and there is quite a lot of literature on this. This stuff is also implemented in code packages, see e.g. the pygformula Python package by the Harvard people to calculate g formula estimates of dynamic treatment strategies. So also on the practical side I am under the impression that we do not have a problem handling these cases.

I have some other points provided below:

Definition 4 seems to be redundant given the text above, I would compress the presentation

wording at line 133 seems to be off: “any” seems to imply that the set of MCV can change the type of any other pair of variables, but your definition just says that a  variable A is meta-causal if there exist two other variables B, C and A can change the type of the relation from B to C. Thus the text and the definition do not seem to say the same thing.

The P in Eq 3 seems to suggest you are thinking of binary outcome, but this was not stated anywhere?

The first paragraph of Section 5 has repetition from the previous paragraph, I would shorten.

Line 306, you mean drug B should be preferred?

On the desirability of outcomes in Section 5.1 and whether drug B is preferable: this seems to be a classic trade-off between short term outcomes (low fever now) and long-term ones (low recurrence of fever in the long run) which will depend on how much the agent discounts the future outcomes and some notion of utility of outcomes. Without setting these parameters, I do not think there is a clear choice of which drug is better unless you unilaterally decide that the short term outcome is less relevant (debatable).

---

> ### Author Rebuttal · Authors · 2025-07-31
>
> Dear Reviewer,
> thank you for your extensive feedback on our paper, pushing us to improve and to provide more explicit contributions. As a direct response to your, but also other reviewers' comments, we have provided a new  algorithm for explicitly estimating (linearized) meta-causal dynamics from a sets of observations. Similar to 'standard' ATE, we furthermore derive a first notion of a meta-causal ATE. We believe that both contributions strengthen our paper and provide a tangible methods for performing meta-causal analyses (MCA). In the following, we will try to answer your remarked points:
>
> 1) **Additional contributions.** First of all, thank you for finding our examples interesting, described and contextualized well, but also for pointing out a lack of actionable contributions and a disconnect between our initial theory and later applications. For the examples we initially aimed to give an intuition of meta-causal aspects, rather than a fully formal treatise. As this point was similarly remarked by other reviewers, we tackle this issue in a two-fold manner: first, we do provide a discussion on the aspects of meta-causal analyses (MCA) together with an explicit algorithm. Second, we provide worked out examples of our applications (utilizing the newly proposed algorithm) to highlight the interplay between classical (time-series) SCM and meta-causal aspects. To keep the discussions compact and easy to parse **we have put the proposed algorithm at the end of this reply and additionally provide a worked out example in our answer to reviewer xvfw.** We will include both in the final paper.
> 2) **Need of new formalism.** We hope our new algorithm and corresponding discussion can clarify the different question being answered in contrast to classical causal (timeseries) models or longitudinal studies. Nonetheless, meta-causal analyses is not a separate analyses methods but designed to be used in combination and to supplement the existing methods. More specifically, MCA considers the transition dynamics between different meta-causal states (/different causal configurations of the system) at a particular points in time. Our newly proposed algorithm, for example, linearizes transition dynamics after the possible unrolling of a time series model to determine transition probabilities between different meta-causal states. (Modeling the applied strategy as either explicit structural equations or continuous **bundles of hard-interventions** is up to the user. However the actual realization is an implementation detail that does not affect the the final analyses. We will put a note in this in the paper). Eventually these state transitions translate to switching structural equations, that, in return, induce particular variable distributions, which can finally be captured via the suggested parametric g-formula, dynamic treatment regime methods and similar statistical analyses. (This is what we had computed so far in our paper). However, MCA is not only interested in the resulting actual effects, but also how such effects come and how states transition into each other with the help of graphical (causal) models, e.g. whether particular mechanisms are active at particular points in time, which might either pose a desired or undesired condition. To better highlight possible questions to be answered by MCA, we partially repeat our answer to reviewer CdGz on naturally arising **question to be answered with MCA**: "How likely is a system to adapt a desired state?", "How stable is a desired state?" or "Which paths (in terms of MCS sequences) are available to reach a particular state?" (and "Are these paths admissible in terms of intermediate system behavior? E.g. are they safe to navigate under safety or ethics standpoints?").
>    To the best of our knowledge, the previous questions can not be answered by existing approaches, as the explicit graphical modeling of qualitative (meta-causal) state transitions is not covered in any of the previously suggested work and constitutes the main novelty/contribution of our paper.
>
> **Other points:**
>
> 1) **Redundant Definition 4.** Agreed. We will remove the explicit equation.
> 2) **Targets of Meta-Causal Variables.** We agree that the presented description can be interpreted ambiguously. What we meant is that possibly every causal edge can be the target of some meta-causal variable (i.e. there is no set of edges that would be inherently protected from being modeled as targeted of a meta-causal variable). We agree that in most cases meta-causal variables influence only a confined fixed set of edge (which, of course, might include influencing the whole set of all edge relations in rare cases). We will be more precise on the nature of this relation in the paper.
> 3) **Eq 3 / Binary Outcome.** You are right. However, this equation is thought to introduce the do-operator for the ATE formalism. We will simply drop the $P$'s here, as they are not required. Kindly, also see our reply to reviewer jRAi on the assumptions required for computing ATEs. We we will state them explicitly in the final version.
> 4) **Repetitions in Sec. 5.** We will reduce the overlap in the camera-ready version.
> 5) **Drug B should be preferred.** Thank you again for catching this mistake on our side. We where switching names of the medications to make the example more easy to follow, but forgot to adjust it in the figure caption. We have corrected it in the paper.
> 6) **On the desirability of outcomes.** We agree. The described setup presents a theoretical scenario with strongly simplified and exaggerated system dynamics, treatment strategies, resulting probabilities and desirability on the purpose of presenting an easy to follow example. In our case, we deem both medications suited to treat all severeness levels of fever. In case this assumption is violated, drug A would need to be administered even in cases of an initial drug B strategy, to avoid severe harm or death. We will add a clarifying disclaimer at the start of the corresponding section to discuss the made assumptions and possible short- versus long-term considerations.
>
> **Summary:** We highlighted similarities and differences of MCM and longitudinal approaches. While we mostly focused on the formal technical details of our work, we are trying to pay attention to the extensive existing work and implications of presenting examples in the fields of general statistics, medicine and epidemiology. Finally, we would like to thank you for pushing us to be more explicit about possible similarities and differences with existing works. This has eventually helped us to better identify out the core contributions of our work. Kindly let us know in case any open points remain.
>
> Best regards,
> the authors
>
> ---
>
> The **Linearized-Meta-Causal-Dynamics (LMCD) Algorithm** estimates the linearized transition probabilities between different meta-causal states from a given set of observed states. For obtaining initial populations at particular timesteps standard causal timeseries SCM can be applied. While existing longitudonal approaches might be sufficient to estimate dynamic treatment effects over several timesteps, the presented algorithm captures the (linearized) state transition probabilities of the system, which reveal the underlying dynamics and paths the system might take:
>
> 1) Input: an SCM $\\mathcal{M} = (\\mathbf{U}, \\mathbf{V}, \\mathbf{F}, P_{\\mathbf{U}})$; a record (or simulated) population of $N$ samples  $\\mathbf{x}^{t} \\in \\mathbf{X}^N, i \\in [1..N]$ at timestep $t$; and an identification function $\\mathcal{I}: \\mathbf{X} \\to T$.
> 2) For every sample $\\mathbf{x}^{i,t} \\in \\mathbf{x}^{t}$:
>    1) Advance the system to another timestep $\\mathbf{x}^{i,t+1} := \\mathbf{F}(\\mathbf{x}^{i,t})$.
>    2) Identify the (possibly coarsened [a]) MCS of the respective states $T^{i,t} := \\mathcal{I}(\\mathbf{x}^{i,t})$ and $T^{i,t+1} := Id(\\mathbf{x}^{i,t+1})$
> 3) Compute the (indexed) set of unique meta-causal states: $U = (\\bigcup_i T^{i,t}) \\cup (\\bigcup_i T^{i,t+1})$ with index $l: T \\to [1..|S|]$
> 4) Compute the entries of the transition probability $P^t \\in \\mathbb{R}^{|S|\\times|S|}$ as $P^t_{u,v} := \\sum_{i\\in[1..N]} (\\delta(l(T^{i,t}) = u) \\land (l(T^{i,t+1}) = v) / \\sum_{i\\in[1..N]} (\\delta(l(T^{i,t}) = v)$ with $u,v \\in [1..|S|]$ and $\\delta$ being 1 if $b$ is true and 0 otherwise.
> 5) [Optional] Compute the continuous time rate matrix via the matrix exponential $Q := e^{P^t-I}$ where $I$ is the identity matrix.
>
> [a] Grouping different $T$ by only considering problem relevant subsets $T'^{i,t} \\subset T^{i,t}$ of the full type matrix.
>
> **Choice of Identification function in the examples:** For all examples and any two continuous variables $X_i, X_j$ connected via a direct causal edge $X_j \\to X_i$, we use the "contextual independency" function $\\mathcal{I}(\\mathbf{x}, X_i, X_j) := (dX_i/dX_j(\\mathbf{x}) \\neq 0)$, which is true (meaning $X_j$ exerts influence $X_i$) if $X_i$ is contextually dependent on $X_j$ (has a non-zero influence / the gradient of $X_j$ onto $X_i$ is non-zero), given the subset of parent values of $X_i$ for current variable configuration $\\mathbf{x}$. Otherwise, it is false, indicating that no current causal influence of $X_j$ on $X_i$ exists.
>
> A **(specific) Meta-Causal ATE (sMCATE)** between two strategies A,B with transitions matrices $P_A, P_B$ (or optionally $Q_A, Q_B$) might be defined as the difference in their transition probabilities $sMCATE_{disc}(P_A,P_B) := P_B - P_A$. Note, that $P_A,P_B$ already represent averages over their respective populations. Contrary to our examples, $P_A$ might represent the no treatment / unintervened environment. The sMCATE might be further compressed into a single number. For particular scenarios, fitting matrix norms might be chosen for this purpose. In the absence of a clear general candidate for defining the general MCATE we abstain from deciding on a definite characterization.

---

> > ### Comment · Reviewer_xvfw · 2025-08-01
> > **Reflection on rebuttal**
> >
> > I want to thanks the authors for the time and substantial amount of effort they put in the rebuttal. The 'Other points' all seem to be addressed satisfactorily.
> > After reading the rebuttal a couple of times and the other review-rebuttal pairs, I would summarize the changes proposed/promised by the authors as i) a new algorithm to estimate meta-causal dynamics, ii) revamped motivation to argue why MCA formalism is needed, iii) new re-work of the flu example, iv) new notion of meta-causal ATE and associated discussion on what assumptions are needed to estimate it and v) a proper comparison with current formalisms.This is on top of all the other changes (e.g. remove Def 4). This looks to me like a whole new paper, or at least a complete overhaul of the submitted one, and goes far beyond the amount of changes allowed from a submission to a camera-ready. I am of the opinion that this newer version would require a new round of peer review, and for this reason I would keep my vote as is (in a sense, the authors seem to agree with me if they also believed so much new content is needed to pad the original submission...).

---

> ### Author Response · Authors · 2025-08-01
>
> Thank you for the comment. We actually completely disagree with the notion that the added content is a completely new paper or we believe that so much content is required to be padded to the original paper. All the provided answers and changes be it the algorithm or a worked out example serve as add-ons to make the paper a bit more clear. This is not to say that the original paper was not clear in our opinion but as always there is always scope for improvement in any paper. We would like to reiterate that the original paper stands on its own and the new content is helpful but does not comprise a completely new paper in our opinion. We would be happy to discuss further.

---

> > ### Comment · Reviewer_xvfw · 2025-08-05
> >
> > Thank you for your reply, I appreciate we might disagree on the point of whether these changes constitute a new paper. Do you however agree with the list of changes I summarized?

---

> > > ### Author Response · Authors · 2025-08-05
> > >
> > > We appreciate your your continued engagement and efforts on our paper. Overall, we agree with your summary of listed changes, but would like to make a few small comments:
> > >
> > > We agree on **i) and ii)**. In addition to the adjustments in the motivation, we will reserve the majority of the additional space in the final paper to add a subsection at the end of Sec. 4 [1], to place the the algorithm and provide a discussion on the assumptions and goals of MCA (**iv**).
> > >
> > > Regarding **iii)**, we will shrink the existing discussion on standard ATE in the examples in favor of providing a more intricate discussion of the meta-causal ATE. Worked-out derivations of MCATE will be discussed in the main text, but calculations be deferred to the appendix.
> > >
> > > For **v)**, we provide a more contrastive discussion between our paper and existing works in the related work section. We will add an overview on existing methods in longitudinal statistics in the main paper (for a possible text, see our rebuttal answer to reviewer jRAi W3/W4) and will provide more detailed considerations as an additional section in the appendix.
> > >
> > > Although discussions gave raise to several key changes in the text, we believe that all of those contributed significantly in clarifying the role of MCA in the context of existing works and to provide a more tangible algorithm and examples. We believe that all of the discussed points significantly strengthened our paper in comparison to the version submitted. Thank you again.
> > >
> > > [1] Possibly cutting the final "Lasting Effects of Interventions" paragraph from the section, due to your mentioned repetitions with the following Sec. 5 and it being subsumed by the discussion on MCA anyway.

---

### Official Review · Reviewer_CdGz · 2025-06-29

**Clarity:** 3
**Significance:** 3
**Originality:** 3
**Rating:** 5
**Confidence:** 3

**Summary:**

This paper presents a strong theoretical framework for reasoning about dynamic causal systems through Meta-Causal Models (MCMs). The motivation — that static ATE-based methods fail when interventions shift causal dynamics — is clear and well stated. The paper demonstrates the ideas through two examples in healthcare and judicial decision-making. The main theoretical contribution is comprehensive, but the paper’s notation, examples, and clarity can be improved to make the work more approachable and practically usable.

**Questions:**

1. Can you clarify the preferred medication in the flu example? Figure 2 and line 306 seem to imply the conclusion is reversed — Medication B appears preferable because it does not suppress the immune system.
2. How are `M_A` and `M_B` determined? More detail would help reproduce the example.
3. Would you consider adding an explicit worked example of an SCM, MCF, MCS, and MCM in the appendix to improve approachability?
4. Related work: Please contrast your contribution more explicitly with prior methods — what exactly is new vs. what exists.
5. Judicial example: You might expand a bit more on how variables/relations change as the intervention (reordering cases) affects the system.
6. How would suggest other researchers build on your paper/framework? Is there, e.g., some code they could get started with (perhaps with sample models/assumptions)?

**Ethical Concerns:**

["NO or VERY MINOR ethics concerns only"]

**Final Justification:**

Key concerns around notation, figure clarity, and related work have been addressed -- as such I have raised my score to accept. Authors have also committed to resolving remaining clarity and reproducibility issues in the final version, which I'm not sure how we can guarantee but it would be good to somehow check, if possible. While the proposed algorithm goes beyond the initial submission, it does not alter the paper’s core contribution -- perhaps it can be suggested to the authors to explicitly say their algorithm is one such instantiation of a formal approach, somehow implying that it wasn't fully vetted.

**Limitations:**

Yes.

**Quality:**

3

**Strengths And Weaknesses:**

### Strengths
- Solid theoretical framework with comprehensive definitions (MCF, MCS, MCM).
- Strong abstract and introduction that motivate the need for meta-causal reasoning.
- Relevant, high-impact application examples illustrate practical value.
- Limitations are honestly acknowledged and related work is comprehensively listed.

### Weaknesses
- Notation could be improved: e.g., lines 324–336 (`f_t/F_t`, `c/c_i/C_i`), and `P_t` is undefined in Sec. 5.2.
- Figure 2 caption and line 306 may be backwards — Medication B should be preferred since it avoids immunosuppression.
- Related work is present but not clearly contrasted — assumes readers infer the differences rather than explicitly discussing them.
- Examples are reasonable but could benefit from more explicit, step-by-step SCM/MCF/MCS/MCM illustrations in the appendix.
- Some readability issues in plots (e.g., Figure 2) — it is hard to tell which lines correspond to what.
- Minor details: consider `\cdot` after `F_t` (line 278, see Appendix A Eq. 10); clarify how `M_A` and `M_B` are obtained (lines 283–284).

---

> ### Author Rebuttal · Authors · 2025-07-31
>
> Dear Reviewer,
> thank you for your thorough review and feedback on our paper. While being limited in response length we will try to comprehensively answer your remarked points:
>
> 1) **W1/Q1: Notation.** The variables (*F*atique, case*C*omplexity and Case*P*ool) of the SCM are defined in Appendix B. However, we admittedly missed to link the corresponding section. We added a link to the full definitions in the appendix and will state variable domains, abbreviations and equations more clearly.
> 2) **W2: Figure 2 Caption.** We switched drug A and B last minute to improve readability. Apparently, we forgot to adjust the figure... We have adjusted the figure caption to now indicate that medication B should be preferred. Thanks for catching that!
> 3) **W3/Q4: Related Work.** In the following we contrast our work against the broader fields mentioned in the paper. For the paper, we will provide more detailed discussions: Unlike traditional causal time series analysis which often assume fixed causal structures, MCM explicitly model transition dynamics between qualitative system configurations. With regard to works on switching causal mechanisms, the novel class of direct MCM strengthens the transition predictions w.r.t. the observed variables, eliminating purely stochastic predictions. Finally, for the vast field of longitudinal statistics, that typically assess (dynamic) average treatment effects over time, the meta-causal analyses specifically focus on the pathways through different meta-causal states, which might yield desirable or undesirable function properties (e.g. stability or active mechanisms).
> 4) **W4/Q3: Step-by-step Illustrations.** Thank you for suggesting a more tangible and step-by-step explanations of our examples and general meta-causal analyses. We will put a full step-by-step on defining all of SCM/MCF/MCS/MCM in the appendix providing a complete start-to-end illustration for setting up meta-causal models. Additionally, we have proposed a novel algorithm (we kindly refer to our reply to reviewer xvfw) and present an actual application of it at the end of this reply.
> 5) **W5: Readability Figure 2.** We will move the SCM of Figure 2 as a subfigure into the text (similar to Figure 3). This will give the remaining plots more space and we will increase label size. Similarly, we will put the medication levels on a separate axis to declutter the plots and visually separate the variables better.
> 6) **W6/Q2: Clarity - cdot and M_A, M_B.** We agree with your suggestions regarding $F_t$. $M_A$ and $M_B$ are the drug specific properties that are given at the end of the first paragraph in Section 5.1. We choose arbitrary, but moderately distinct values of for the sake of giving an example that is easy to follow. Starting conditions in the Flu example are time=0 and immuneStrength=2. All other values follow from them together with random sampling exposure levels. We will note this in the appendix.
> 7) **Q5: Judicial Example.** Full equations for the example are presented in Appendix A. Nonetheless, we will add a brief description within the main text. Basically, all structural equations in the system remain fixed for the whole scenario. Biased decision arise from increasing fatigue (which simply increases over time) in combination with complex cases. While equations stay the same, some of these relations might only exhibit *actual* effects (e.g. they propagate non-zero values) in certain contexts. Particularly, we are interest in avoiding decision bias. For this we need to prevent the decision bias relation to become active (-> preventing it to compute a non-zero value). As the activation function of the Decision Bias is a ReLu (max(Fatigue+CaseComplexity-5, 0)) we can identify inactive states by having no gradients for $F$ and $C$, in all cases of $F+C<5$ is less than the threshold (5). As fatigue is deterministic, but can not be controlled,  the only chance to avoid the function activation is to implement a case scheduling strategy that selects complex cases at earlier timesteps, and keeps the simple cases for the later timesteps. While classical causal analysis might come to the same conclusions w.r.t. choosing the preferred strategy, the meta-causal approach is purely motivated via actual effect propagation. No further metrics in terms of ATE or similar need to be specified by the user.
> 8) **Q6. Other researchers build on your paper/framework.** Thank you for the interesting question. As already discussed earlier, and given your feedback and that of the other reviewers, we provide an explicit algorithm in our response to reviewer xvfw. We hope our newly provided approach enables meta-causal analyses (MCA) to be out-of-the box applied to other scenarios and make the whole topic of meta-causality more tangible. Similarly, in the answer W3/W4 to reviewer jRAi we elaborated further on the distinction of 'classical' dynamical analyses and meta-causal analyses (MCA), in that are concerned in the change of qualitative system dynamics. Natural question to answer with MCA would be "How likely is a system to adapt a desired state?", "How stable is a desired state?" or "Which paths (in terms of MCS sequences) are available to reach a particular state?" (and "Are these paths admissible in terms of intermediate system behavior? E.g. are they safe to navigate under safety or ethics standpoints?"). Such questions might, for example, arise in the course of planning actions of autonomous/agentic agents, or might be relevant for general societal or governmental decision making.
>
>
>
> Thank you once again for pointing out several unclarities of our work. We believe that our planned changes will overall strengthen the paper in terms of contributions and clarity. We happy to answer any remaining questions or engage in further discussions.
>
> Best regards,
> the authors
>
>
>
> ---
>
> **Worked out example: Medicating Flu.**
>
> As already mentioned for the LMCD algorithm, we chose an identification function that measures *actual* influence between variables ($\\mathcal{I}(\\mathbf{x}, X_i,X_j) := (dX_i/dX_j(\\mathbf{x}) \\neq 0)$). The function is true whenever $X_i$ is contextually dependent on $X_j$ (<=> the gradient of $X_j$ on $X_i$ is non-zero) and false otherwise. By identifying the full meta-causal types of all recorded samples, we obtain 10 unique MCS for drug A and 8 unique MCS for drug B. For this analysis, we however only focus on the relevant activations of edges that affect body temperature. This coarsens the MCS into 3 unique states with either none of the edges active (1),  {Fever->Temperature} active (2) or {Fever->Temperature, Medication->Temperature} active (3). (Note that {Medication->Temperature} does never occur, since medication is only administered in case of fever).
>
> By counting the occurrence of individual meta-causal states at timestep $t=9$ ("age 18") we get the following vectors:
>
> $C^t_A = \\begin{bmatrix} 10 \\\\ 9 \\\\ 981 \\end{bmatrix};\\quad C^t_B = \\begin{bmatrix} 437 \\\\ 104 \\\\ 459 \\end{bmatrix}$
>
> Advancing individual states one step further, we obtain the following absolute transition counts:
>
> $C_A = \\begin{bmatrix} 3 & 0 & 7 \\\\ 1 & 1 & 7 \\\\ 0 & 6 & 975 \\end{bmatrix}; \\quad C_B = \\begin{bmatrix} 227 & 40 & 170 \\\\ 33 & 28 & 43 \\\\ 120 & 58 & 281 \\end{bmatrix}$
>
> The per-row normalized transition matrices then model the state changes probabilities at time $t$:
>
> $P_A = \\begin{bmatrix} 30.00\\% & 0.00\\% & 70.00\\% \\\\ 11.11\\% & 11.11\\% & 77.78\\% \\\\ 0.00\\% & 0.61\\% & 99.39\\% \\end{bmatrix}$
>
>  $P_B = \\begin{bmatrix} 51.95\\% & 9.15\\% & 38.90\\% \\\\ 31.73\\% & 26.92\\% & 41.35\\% \\\\ 26.14\\% & 12.64\\% & 61.22\\% \\end{bmatrix}$
>
> And similar continuous-time rate matrix: (for discrete time steps $P$ might be sufficient.)
>
> $$Q_A = \\begin{bmatrix} 0.496 & 0.001 & 0.502 \\\\ 0.050 & 0.412 & 0.537 \\\\ 0.000 & 0.004 & 0.995 \\end{bmatrix}$$
>
>  $$Q_B = \\begin{bmatrix} 0.662 & 0.066 & 0.271 \\\\ 0.211 & 0.506 & 0.282 \\\\ 0.186 & 0.082 & 0.731 \\end{bmatrix}$$
>
> Specific Meta-Causal ATE (effect between both medication; more commonly effect between treatment and no treatment might be considered):
>
> $sMCATE_{disc}(P_A,P_B) = P_B - P_A = \\begin{bmatrix} 21.94\\% & 9.15\\% & -31.09\\% \\\\ 20.61\\% & 15.81\\% & -36.43\\% \\\\ 26.14\\% & 12.02\\% & -38.16\\% \\end{bmatrix}$
>
> **Interpretation:** From the $C$ matrices we see that the third state, indicating fever induced medication, is almost always active for drug A (98.1%), while it is only given in 45,9% of the cases for drug B. Considering the meta-causal transition matrices $P_A, P_B$, we do not only find that the MCS of fever induced medication is increased in A, but also that all other states eventually flow into this state (with probabilities of 70,0% and 77.78%). Drug B features more favorable dynamics. Here, patients with a fever induced medication are still rather likely to to stay in that state. However, (moderately) healthy patients only transition into this state with moderate probabilities of 38.9% and 41.35% and are similarly able to leave leave it. These differences in dynamics are also visible from considering the sMCATE matrix, which shows a sharp decrease in transition probabilities into the third state, while the healthy state observes a strong increase in incoming transition probabilities.
>
> **Additional notes:** The computed $P$ and $Q$ matrices can be interpreted as Markov and continuous flow processes. While this years reviewing process does not allow to update materials, we will include corresponding graph visualizations in the paper.
>
> **Judicial Decision-Making.** While we had to skip the judicial example due to the character limit of the initial response, we will likewise provide the corresponding calculations in the final paper which follow the same algorithm steps. Since outcomes in this scenario are deterministic for the optimal strategy, we deemed the medicating flu example to be more worthwhile to present in our response.

---

> > ### Author Response · Authors · 2025-08-04
> > **Any further concerns?**
> >
> > Dear reviewer,
> >
> > Since the discussion period will end in a couple of days, we would like to ask if there are any further questions from your side. We have replied in detail to all your original concerns and hope that we have alleviated your concens. We would be happy to discuss further if necessary.
> >
> > Regards,
> >
> > The authors

---

> > > ### Comment · Reviewer_CdGz · 2025-08-05
> > > **Official Comment (score increased)**
> > >
> > > Thank you for the detailed rebuttal -- you have addressed my immediate concerns (W1/Q1 on notation, W2 on Figure 2, and W4/Q3 on related work) directly in your response, and promised revisions to further clarify the remaining points I raised (W3/Q4 on worked examples and W6/Q2 on model clarity and derivation). As such, I am increasing my score to accept.
> > >
> > > Regarding your newly proposed algorithm and the meta-causal ATE formulation: while I agree with the other reviewers that such contributions might typically warrant a separate paper and full round of peer review, I recognize the intention here is to supplement and clarify—not to pivot the core thesis of the original submission. Given that peer review should ultimately aim to help authors improve and disseminate impactful work in a timely manner, I read through the new material and found it directionally promising, though not fully vetted. A thorough evaluation (e.g., benchmarking, limitations, comparative baselines, theoretical tradeoffs) would indeed merit its own space and possibly its own paper.
> > >
> > > That said, I would strongly re-encourage you to include a worked example (also for this algorithm)—even if placed in the appendix but clearly referenced from the main body—to illustrate the pipeline from SCM → MCF/MCS/MCM → analysis. This would make your work much more approachable for readers across different backgrounds.
> > >
> > > Overall, I enjoyed reading your paper and I look forward to seeing and sharing the final version around.

---

> > > > ### Author Response · Authors · 2025-08-06
> > > >
> > > > Dear reviewer,
> > > > Thank you for the fruitful discussion and encouraging perspective on our paper. We have already started revising the paper and will make sure that all details related to SCM/MCF/MCM construction and the follow-up analysis, currently scattered around the paper and our various replies, will be provided as a fully worked example in the appendix.

---

### Decision · Program_Chairs · 2025-09-17

**Decision:**

Accept (poster)

**Comment:**

This submission builds on previous work about meta causal models (MCMs) that allow for causal structure to be dynamically updated, resulting in the same intervention having different effects at different points in time. The strengths of the paper are that it is clearly written, well motivated, and has examples of how the theoretical framework can be put into action. The main weakness of the paper identified by Reviewer xvfw is that there is only one theorem with a short proof and the examples don't use the new formalism, and so the contributions of this paper are light. There was a vigorous discussion between reviewers and authors, and several pieces of new material were proposed by the authors, which the reviewers agreed are high quality, but Reviewer xvfw felt that the resulting changes would be too great.


Ultimately, my decision to suggest this paper be accepted is because the new material proposed during discussion with reviewers was given at a high level of detail with mathematical rigor. While we can't have a perfect picture of how the author will incorporate the material, the reviewers seem confident that the new material is of high quality and improves the submission. My inclination is to reward such a lively and productive discussion!